# An integrated map of fibroblastic populations in human colon mucosa and cancer tissues

Siying Li[1,7], Ran Lu[1,2,3,7], Linjuan Shu[1], Yulin Chen[1], Jin Zhao[1], Junlong Dai[4], Qiaorong Huang[1], Xue Li[1], Wentong Meng[1], Feiwu Long[5], Yuan Li[6], Chuanwen Fan [5,6✉], Zongguang Zhou [6✉] & Xianming Mo [1✉]

Fibroblasts and myofibroblasts are major mesenchymal cells in the lamina propria of colon mucosa and in colon cancer tissues. Detailed insight into the highly specific populations of fibroblasts and myofibroblasts is required to understand the integrity and homeostasis of human colon mucosa and colon cancer. Based on gene expression profiles of single cells, we identified fibroblast populations that produce extracellular matrix components, Wnt ligand- and BMP-secreting fibroblasts, chemokine- and chemokine ligand-generating fibroblasts, highly activated fibroblasts, immune-modulating fibroblasts, epithelial cell-modulating myo-fibroblasts, stimuli-responsive myofibroblasts, proliferating myofibroblasts, fibroblast-like myofibroblasts, matrix producing myofibroblasts, and contractile myofibroblasts in human colon mucosa. In colon cancer tissue, the compositions of fibroblasts and myofibroblasts were highly altered, as were the expressing patterns of genes including BMPs, Wnt ligands, chemokines, chemokine ligands, growth factors and extracellular matrix components in fibroblasts and myofibroblasts. Our work expands the working atlas of fibroblasts and myofibroblasts and provides a framework for interrogating the complexity of stromal cells in human healthy colon mucosa and colon cancer tissues.

[1] Laboratory of Stem Cell Biology, State Key Laboratory of Biotherapy, West China Hospital, Sichuan University, Chengdu, China. [2] Department of Public Health Laboratory Sciences, West China Fourth Hospital, West China School of Public Health, Sichuan University, Chengdu, China. [3] Department of Urology and Pelvic Surgery, West China Fourth Hospital, West China School of Public Health, Sichuan University, Chengdu, China. [4] Department of Liver Surgery, West China Hospital, Sichuan University, Chengdu, China. [5] Department of Gastrointestinal, Bariatric and Metabolic Surgery, Research Center for Nutrition, Metabolism & Food Safety, West China-PUMC C.C. Chen Institute of Health, West China Fourth Hospital, West China School of Public Health, Sichuan University, Chengdu, China. [6] Institute of Digestive Surgery and Department of Gastrointestinal Surgery, West China Hospital, Sichuan University, Chengdu, China. [7] These authors contributed equally: Siying Li, Ran Lu. ✉email: xuntian2005@163.com; Zhou767@163.com; xmingmo@scu.edu.cn

Fibroblasts, which synthesize the extracellular matrix and structural proteins such as collagen and elastin in mesenchymal tissue, are α-SMA negative and are located adjacent to the myofibroblasts surrounding colonic crypts in the lamina propria of colon mucosa[1,2]. Previous observations have shown that myofibroblasts, which share morphological features with conventional tissue fibroblasts and contractile smooth muscle cells[3], are present in the pericryptal fibroblastic sheath in the colon mucosa[4]. Myofibroblasts are spindle-shaped cells characterized as α-SMA+, vimentin+, non-smooth muscle myosin+, fibronectin+ and desmin- [1,5]. Both fibroblasts and myofibroblasts coordinate diverse functions in the colon in health and disease and play critical roles in homeostasis maintenance, epithelial growth and repair, immune response, inflammation, fibrosis, tumorigenesis, and cancer progression. For instance, in colitis and inflammatory bowel disease, dysfunction of fibroblast subsets impairs epithelial proliferation and maturation and contributes to disease severity[6–8]. In the lamina propria of murine colon mucosa, two functionally distinct fibroblast populations have been identified[9]. Crypt-top fibroblasts, located at the top of crypt, express a high level of PDGFRα and secrete noncanonical Wnt ligands and Bmp ligands, thereby inducing epithelial differentiation. The PDGFRα$^{low}$ crypt-bottom fibroblasts, located at the bottom of the crypts in close proximity to epithelial stem cells, secrete canonical Wnt ligands, Wnt signaling potentiators and Bmp inhibitors and might contribute to the maintenance of the stem cell niche. The human colon mucosa is also found to have such two populations with diverging gene expression patterns[9].

Fibroblasts and myofibroblasts are also the major stromal cells in cancer tissues. Fibroblasts are a dominant stromal component in cancer and have been termed cancer-associated fibroblasts (CAFs)[10,11]. CAFs are activated fibroblasts and have been shown to be key components in tumor stroma programming and support cancer progression and metastasis[11]. Cancer-associated myofibroblasts are also activated and play complex and bimodal roles in cancer promotion, cancer progression, and metastasis[12–14]. Fibroblasts and myofibroblasts in the cancer stroma secrete extracellular matrix components, cytokines, chemokines, and growth factors to promote cancer cell survival and migration, regulate the constitution of the matrix to facilitate cancer progression, alter immune responses to allow cancer cells to evade immune surveillance, and reprogram the cancer microenvironment to adapt to and develop resistance to therapy[10–12]. In patient-derived xenografts, high expression of the CAF signature molecules is associated with poor prognosis, and mutual high expression of the stromal signatures predicts resistance to therapy in colorectal cancer[14,15]. Individual cell assays have revealed that non-small-cell lung cancer tissues contain five distinct types of fibroblasts and myofibroblasts[16]. In bladder urothelial carcinoma, there are two types of fibroblasts[17]. One of them is the RGS5+ fibroblast, which has characteristics similar to those of cancer-associated myofibroblasts. Another population is the PDGFRα+ fibroblast, which exhibits strong expression of various cytokines and chemokines, similar to the inflammatory cancer-associated fibroblasts identified in a pancreatic cancer model[18]. CD105+ pancreatic fibroblasts have been shown to promote tumor growth, and CD105- fibroblasts are highly tumor suppressive in murine models[19]. In breast cancer tissue, myofibroblasts, characterized by extracellular matrix proteins or TGFβ signaling, are indicative of primary resistance to immunotherapies[20]. Human triple-negative breast cancer tissue contains myofibroblasts and inflammatory fibroblasts characterized by high expression of growth factors and immunomodulatory molecules[21]. Inflammatory fibroblasts in cancer patients have been revealed strong associations with cytotoxic T-cell dysfunction[21,22]. In pancreatic ductal adenocarcinoma, LRRC15+

CAFs correlated with poor response to anti-PD-L1 therapy, and a subtype of fibroblasts identified expressing MHC class II and CD74 act as antigen-presenting cells[23,24].

Thus, myofibroblasts and fibroblasts are complex and heterogeneous cells that perform diverse functions in the lamina propria of colon mucosa and in colon cancer tissues. Highly specific populations are required further to reveal deeper mechanistic insights into fibroblasts and myofibroblasts in the lamina propria of human colon mucosa and colon cancer tissues to understand how the integrity and homeostasis of colon mucosa are maintained under diverse physiological and pathological conditions and how cancer cells survive and develop. Here, we measured fibroblasts and myofibroblasts from paired samples of colon cancer tissue and the adjacent normal colon mucosa of each subject by single-cell RNA-sequencing. The results provide a detailed atlas of fibroblasts and myofibroblasts in human healthy colon mucosa and colon cancer tissues. Moreover, the data show that there are distinct types of fibroblasts and myofibroblasts in human colon mucosa and that the types of fibroblasts and myofibroblasts are altered in colon cancer tissues.

## Results

**Unsupervised single-cell profiling of fibroblasts and myofibroblasts in human colon mucosa.** To draw up a specific atlas of fibroblasts and myofibroblasts in normal human colon mucosa and colon cancer tissues, we collected paired samples of human colon cancer tissues and the adjacent normal colon mucosa from 12 patients and dissociated each sample into single cells to generated single cell labelled cDNA libraries for sequencing. The colon mucosa and colon cancer cell suspensions from 9 patients were sorted by fluorescence-activated cell sorting (FACS) to enrich CD326-, CD45- and CD31- cell populations (Supplementary Fig. 1a) and used to generate single cell cDNA libraries with BD Rhapsody system for sequencing. The CD45- sorted colon mucosa cells and colon cancer cells from additional three pairs of specimens were generated single cell labelled cDNA libraries with Singleron GEXSCOPE for sequencing. The cell populations were categorized by data mining (Supplementary Fig. 1b, c). Both single-cell RNA-sequencing datasets are highly matched. Combined pre-analysis of all cells, as predicted, in the datasets multiple cell types were identified, including T cells, B/Plasma cells, epithelial cells, gut Gila cells, and dendritic cells, endothelial cells, fibroblastic cell populations, and a small fraction of mast cells (Supplementary Fig. 1d, e).

After exclusion of the non-fibroblastic cell populations, 8935 fibroblastic cells from the samples of human colon mucosa were analyzed. After correction for technical and biological variation and quality control, 11 individual clusters, which can be classified into fibroblasts and myofibroblasts using known markers[8], were established based on the patterns of 3000 HVGs expression profiles (Fig. 1a, b, Supplementary Data 1, 2). Each cluster was found in the normal mucosa of several patients (Fig. 1c, Supplementary Data 1). All clusters expressed *VIM*, *S100A4*, and *COL1A1*. The additional classic fibroblastic marker genes such as *DCN*, *ACTA2*, *FAP* and *PDGFRA* were highly expressed in several subsets (Fig. 1d, f). According to the similarity of marker gene expression patterns, the clusters could be grouped into three populations, including OGN-expressing, CXCL14-expressing fibroblasts, and RGS5-expressing myofibroblasts (Fig. 1e, f, g). Two clusters of a small fraction of cells with distinct gene expression in the CXCL14-expressing fibroblastic cells, HHIP+ myofibroblasts and CCL19+ fibroblasts were identified. Each fibroblastic subpopulation had specific gene expression profiles, especially in extracellular matrix, cytokines, and growth factors (Fig. 2a, b, c), suggesting that each subpopulation might display different biological functions

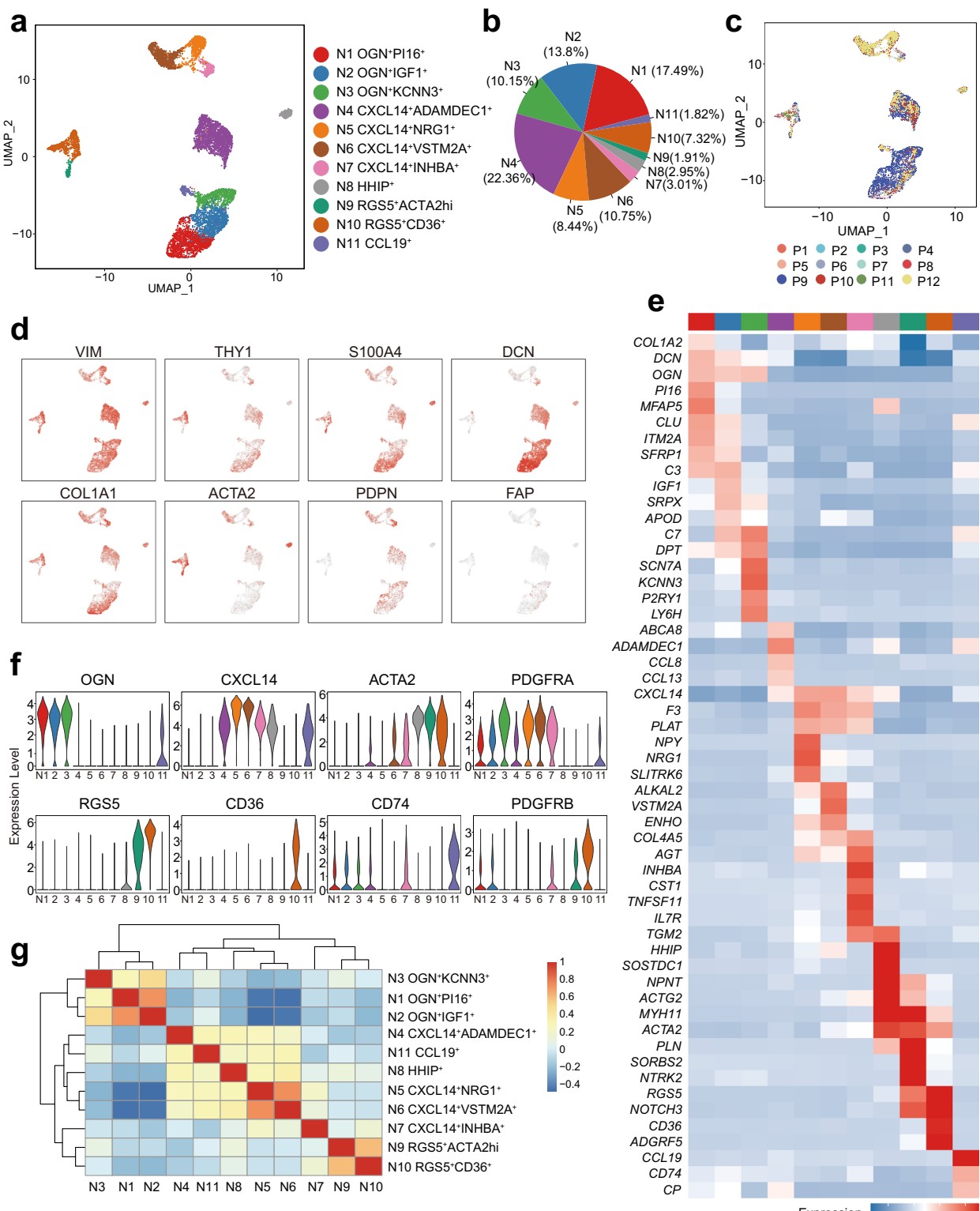

**Fig. 1 scRNA-seq analysis defined types of fibroblasts and myofibroblasts in human colon mucosa. a** UMAP projection of fibroblasts from normal colon mucosa, colored by inferred cluster identity. **b** Proportions of the clusters in all fibroblasts. **c** UMAP projection of normal colon mucosa fibroblasts colored according to the patients (labeled P1 to P12, $n = 12$). **d** UMAP Feature plots of the expression distribution of classical fibroblast markers**. e** Heatmap showing the scaled expression of marker genes in each cluster of mucosal fibroblasts. **f** Violin plots showing selected distinct markers of clusters. **g** Heatmap of Pearson correlation coefficients between the normal colon mucosa fibroblast clusters. The dendrogram shows the inferred hierarchy of clusters according to the Pearson correlation coefficients of the average expression of marker genes among clusters. The color scale shows the Pearson correlation coefficient.

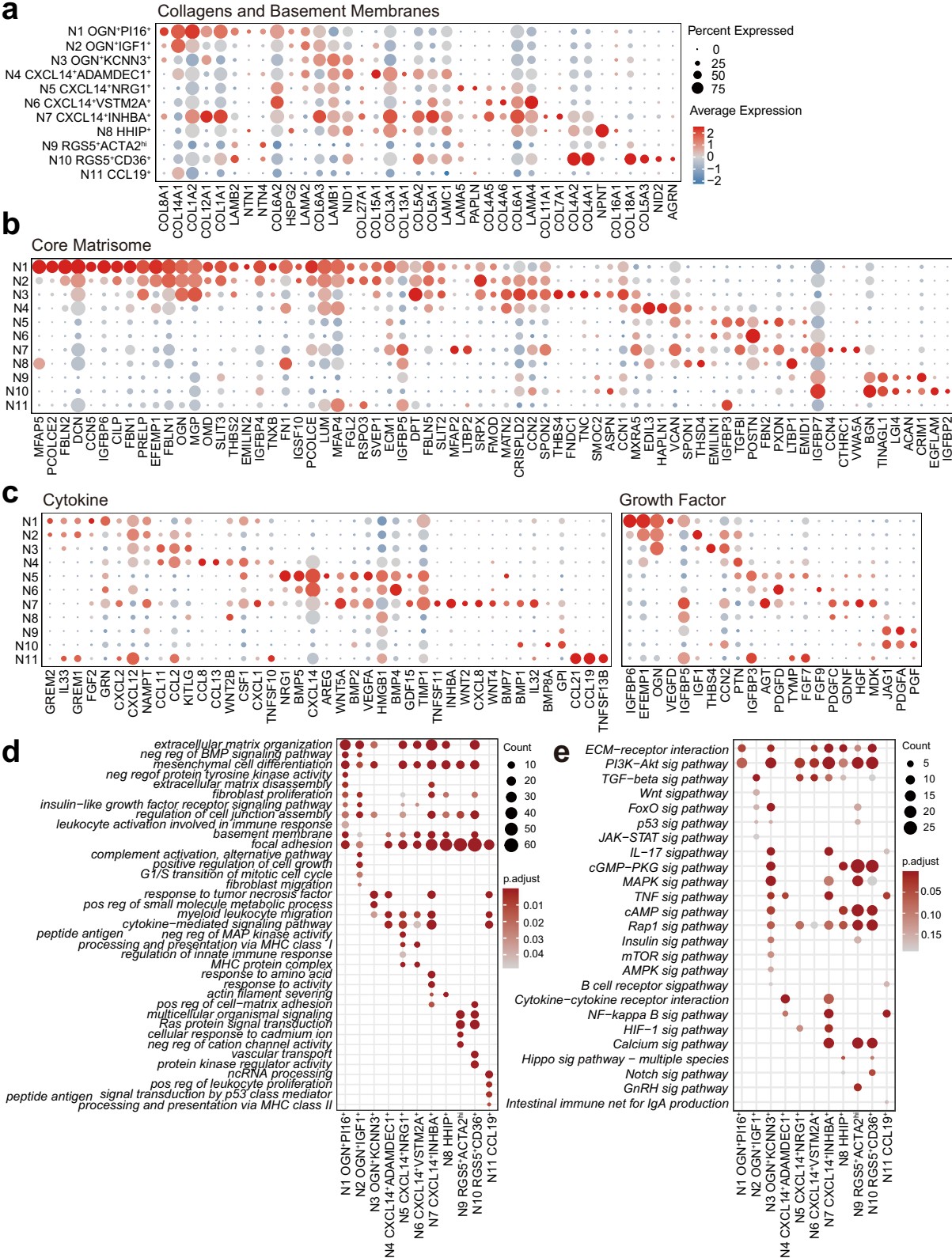

(Fig. 2d, e). Taken together, these data represent a comprehensive, high-quality dataset demonstrating the cell diversity of fibroblasts and myofibroblasts in human colon mucosa.

**Population connectivity of fibroblasts and myofibroblasts in human colon mucosa**. The transcriptome expression data obtained from the large number of fibroblasts and myofibroblasts

in the human colon mucosa allowed us to obtain insights into the functional and constitutive relationships between these cells. We explored the relationships and connectivity among the fibroblasts and myofibroblasts by inferring the state trajectory using Monocle[25]. The trajectory analysis of more than 3000 gene expression patterns showed that myofibroblasts and fibroblasts were aggregated into five cellular states (Fig. 3a). Specifically, we

**Fig. 2 The patterns of gene expression profiles in human colonic mucosa fibroblasts and myofibroblasts. a–c** Dot plot showing the scaled expression of several key biological function-related gene sets of each cluster in mucosa. Gene sets were downloaded from the Molecular Signatures Database (MSigDB). The color gradient represents the average expression of each gene in each cluster scaled across all clusters. The dot size represents the percentage of positive cells in each cluster. The colors representing each cluster in the color band correspond to those in Fig. 1a. **a** Genes encoding structural components of basement membranes (M5887) and collagen proteins (M3005). **b** Genes encoding core extracellular matrix components, including ECM glycoproteins, collagens and proteoglycans (M5884), excluding the genes contained in Fig. 4A. **c** Genes encoding cytokines (GO:0005125) and growth factors (GO:0008083). **d** Gene Ontology analysis of marker genes for each cluster. **e** KEGG enrichment analysis of marker genes for each cluster.

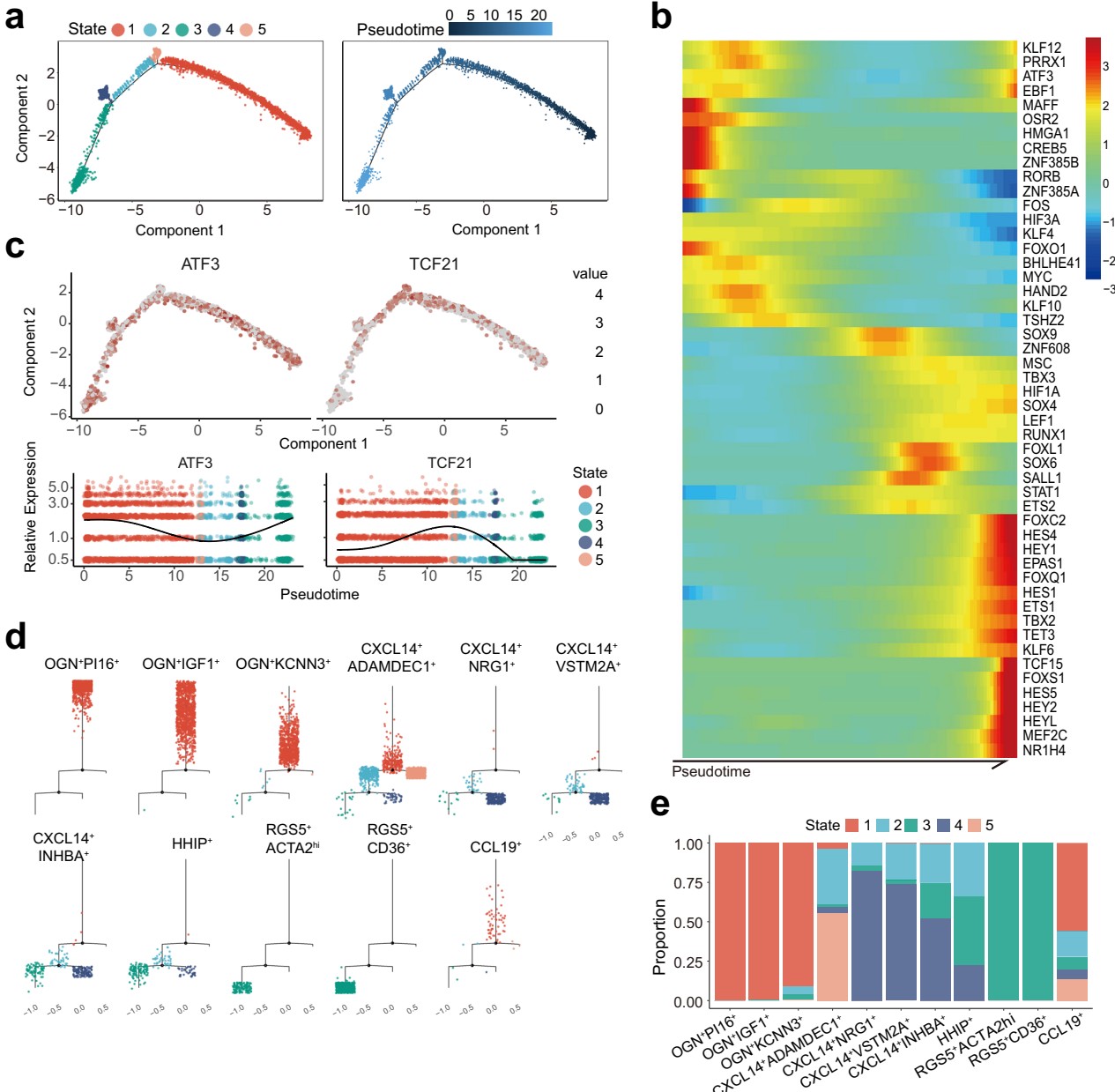

**Fig. 3 Trajectory analysis of fibroblasts and myofibroblasts in human colonic mucosa. a** Monocle trajectories of fibroblasts from normal mucosa. Cells are colored by Monocle state (left) and pseudotime (right). **b** Heatmap showing the dynamic changes in pseudotime-dependent transcription factors over pseudotime. **c** Feature plots of the expression distribution for selected transcription factors across pseudotime. **d** The stacked bar chart shows the percentage of fibroblasts from distinct clusters across the pseudotime states. **e** Plots of the minimum spanning tree on cells, split by cluster.

paid much attention to the expression of transcription factors, which display essential roles on the cell states and identities. The results of the expressing analysis showed dynamic expression change of a series of transcription factors, e.g., *ATF3* and *TCF21*,

over pseudotime courses (Fig. 3b, c), suggesting that the trajectory analysis is reliable.

The three types of fibroblasts with *OGN* expression were found to be ordered into the same functional and constitutive group and were

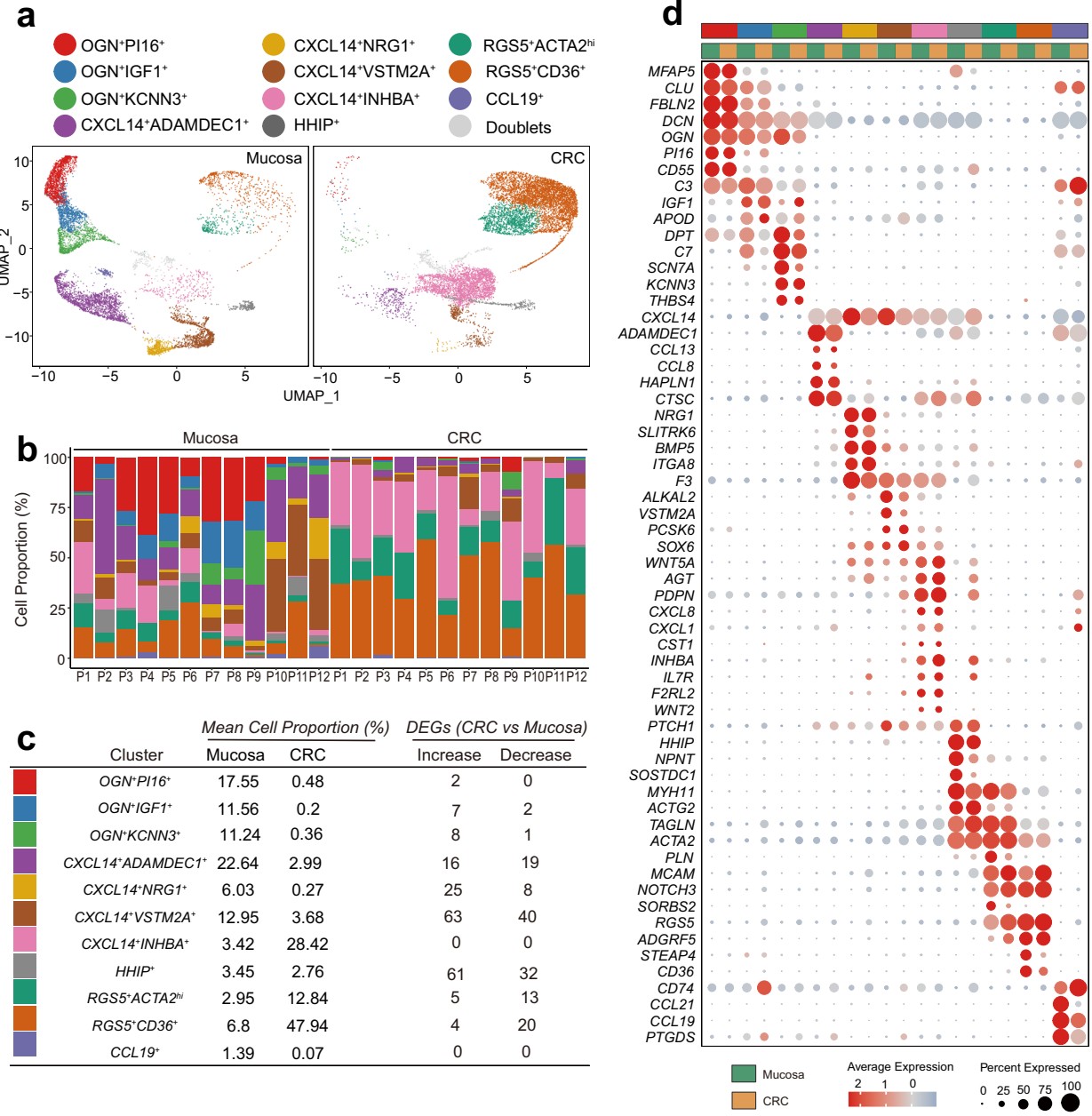

**Fig. 4 Integrated analysis of fibroblasts and myofibroblasts in human colonic mucosa and in human colon cancer tissues. a** Comparison of colonic mucosal fibroblasts and colon cancer fibroblasts. UMAP projection of integrated analysis displayed separately by sample type. **b** Comparison of cell composition between mucosa and cancer tissues in each patient. **c** Table displays comparison of mean cell proportion of each cluster between mucosa and cancer tissue and number of significantly differentially expressed genes (padj < 0.05 & Log$_2$FC >1, decrease Increase: padj < 0.05 & Log$_2$FC < -1). **d** Dot plot showing the scaled expression of signature genes for each cluster in mucosa and CRC. The colors representing each cluster in the color band correspond to those in Fig. 4a.

clustered as an independent group in the trajectory (Fig. 3a, d, e). OGN$^+$PI16$^+$ fibroblasts displayed steady state and were clustered independently from the other groups of myofibroblasts and fibroblasts in human colon mucosa. RGS5$^+$ myofibroblasts were distributed in state 3, one group in the trajectory. CXCL14-expressing populations were arranged between OGN$^+$ fibroblasts and RGS5$^+$ myofibroblasts, and they had two relatively stable differentiation states. CXCL14$^+$ADAMDEC$^+$ fibroblasts exhibited some similarity to OGN-expressing fibroblasts and the majority of cells were in a steady state. As for the remaining CXCL14$^+$ populations including CXCL14$^+$NRG1$^+$, CXCL14$^+$VSTM2A$^+$, CXCL14$^+$INHBA$^+$, and HHIP$^+$ fibroblasts and myofibroblasts, fractions of cells were in a transitional state, and others were in distinguishable states (Fig. 3a, d).

CXCL14$^+$INHBA$^+$ population was close to RGS5$^+$ myofibroblast in the expression profiles and Pseudotime series (Fig. 1g, Fig. 3a), suggesting that there may be a transformational relationship between them. However, the relationship and the patterns of mutual variation among these subpopulations may not be distinctive.

**Fibroblasts and myofibroblasts in colon cancer tissues.** Next, 10,986 cells from the 12 samples of human colon cancer tissues were subjected to analysis. In integrated analysis, the populations of cancer-related myofibroblasts and fibroblasts were found to be highly similar to the populations of myofibroblasts and fibroblasts in human colon mucosa (Fig. 4a, b, Supplementary Data 3, 4). The dominant populations of cancer-related myofibroblasts and

fibroblasts include RGS5-expressing myofibroblasts and CXCL14-expressing fibroblasts (Fig. 4b, c). A small fraction of cells aggregated into OGN[+] fibroblast populations, indicating that OGN[+] fibroblast populations dramatically decreased in cancer tissues. Gene profiling of single cells revealed that RGS5-expressing myofibroblasts, CXCL14-expressing fibroblasts and HHIP[+] myofibroblasts showed extensive upregulation of extracellular matrix genes, including collagen genes, in colon cancer tissues (Fig. 4c, d, Supplementary Fig. 2a, b, Supplementary Data 5). In addition, the fibroblasts and myofibroblasts in cancer tissues were altered to express increasing levels of a panel of genes, including BMP, Wnt ligands, chemokines, chemokine ligands, and growth factors (Supplementary Fig. 2c, d). The results suggest that fibroblasts and myofibroblasts in cancer tissues are active constituents of stromal tissues which support the expansion of cancer and to modulate inflammatory reactions and cell proliferation in cancer tissues.

To verify the types of fibroblasts and myofibroblasts, we employed anti-human CD31, CDX2, Cytokeratin 20, and CD45 antibodies to stain endothelial cells of blood vessels, inflammatory cells and epithelial cells or cancer cells in the tissues of samples. The fibroblasts and myofibroblasts were visualized by anti-human RGS5, decorin, podoplanin/gp36, CD36, α-SMA, and HHIP antibodies in colon mucosa (Fig. 5a–h) and colon cancer tissues (Fig. 6a–k).

**Myofibroblasts in human colon mucosa and colon cancer tissues.** There were three types of myofibroblasts in human colon mucosa and in colon cancer tissues. Among them, the HHIP[+] myofibroblasts expressed both *ACTA2* and the chemokine *CXCL14* and specifically expressed hedgehog-interacting protein (HHIP) (Fig. 1e, Fig. 5a, Supplementary Fig. 3a, b). These myofibroblasts also expressed the canonical WNT ligands WNT2B and SOSTDC1, a soluble BMP and Wnt antagonist (Fig. 1e, Fig. 4d). Immunostaining showed that HHIP-expressing myofibroblasts were sparsely located at the bottom of colon crypts (Fig. 5a) in the human colon mucosa. The proportion of HHIP[+] myofibroblasts was not altered in colon cancer tissue (Supplementary Fig. 3a) compared to human colon mucosa. However, gene activation in cancer-related HHIP[+] myofibroblasts was highly altered (Supplementary Fig. 2, Supplementary Fig. 3c, d, e). The expression of the noncanonical WNT ligand WNT5A, many chemokines, and many chemokine ligands was increased. However, the expression of the canonical WNT ligand WNT2B was not affected. The expression patterns of transcription factors and members of several hallmark signaling pathways were also altered (Supplementary Fig. 2e, f, Supplementary Fig. 3e). For instance, the expression of genes involved in signaling pathways that regulate the immune response, including T-cell differentiation, was greatly increased (Supplementary Fig. 3c, d). The results suggest that HHIP[+] myofibroblasts might adapt and respond to regulate cancer cell proliferation and to modulate immune reactions in cancer tissues. Consistently, HHIP staining showed that the myofibroblasts were sparsely distributed and located close to cancer cells (Fig. 6a).

The other myofibroblasts in the human colon mucosa expressed *RGS5*. Based on the expression of *CD36* and *ACTA2*, RGS5-expressing myofibroblasts were clustered into two types (Fig. 1a). One type of myofibroblast was RGS5[+]ACTA2[hi] cells with higher expression of α-SMA, and the other type of cells was RGS5[+]CD36[+] cells with lower expression of *ACTA2*. Previously, RGS5[+]ACTA2[+] cells were identified as pericytes[8,26], the mural cells of blood microvessels. In our data, RGS5[+]ACTA2[hi] myofibroblasts were not found to express markers of pericytes[1] (Fig. 4d). In contrast, RGS5[+]CD36[+] myofibroblasts carried some markers for pericytes (Fig. 1f, Fig. 4d). For example, RGS5[+]CD36[+] myofibroblasts expressed PDGFRβ, but lacked Desmin expression, suggesting that the cells were more likely to be myofibroblasts. Indeed, RGS5 and CD36 staining confirmed that both RGS5[+]CD36[+] and RGS5[+]ACTA2[hi] cells were myofibroblasts located along the crypts in human colon mucosa (Supplementary Fig. 4a). α-SMA staining showed that RGS5[+]CD36[+], RGS5[+]ACTA2[hi], and HHIP[+] myofibroblasts were fusiform-shaped cells immediately subjacent to epithelial cells, encircling colonic crypts just under colonic epithelial cells and consisting of subepithelial stromal cells (Fig. 5a, b, c, d, Supplementary Fig. 4b, c).

In a detailed analysis, the RGS5[+] myofibroblasts were then grouped into seven subpopulations (Supplementary Fig. 5a, b, c, Supplementary Data 6, 7). RGS5[+]CD36[+]cells were divided into three subpopulations of myofibroblasts, ACTA2[low]CD248[hi], ACTA2[low]PFKFB3[hi], and CD36[hi] myofibroblasts. The ACTA2[low]CD248[hi], ACTA2[low]PFKFB3[hi] populations was identified to highly express a panel of collagen genes (Supplementary Fig. 5c). ACTA2[low]PFKFB3[hi] population was involved in phosphatase activity and fructose and mannose metabolism. The gene expression patterns in CD36[hi] myofibroblasts were shown to be involved in several stimulus-responsive pathways, including cytokine-responsive pathways, immune response-regulating signaling pathway, and participated in fatty acid transport (Supplementary Fig. 5c), indicating that CD36[hi] myofibroblasts might respond to stimulations in tissue microenvironments.

The RGS5[+]ACTA2[hi] populations (ACTA2[hi]S100A4[hi] and PDK4[+]) expressed high levels of contractile apparatus components, including the actin binding proteins destrin (DSTN) and transgelin (TAGLN), the myosin heavy chain protein MYH11, and the myosin light chain regulatory subunit MYL9 (Supplementary Fig. 5c). A small fraction of RGS5[+]ACTA2[hi] population (PDK4[+]) highly expressed Phospholamban (PLN) and Pyruvate Dehydrogenase Kinase 4 (PDK4) and contributed to stress response to metal ion and cellular response to hydrogen peroxide (Supplementary Fig. 5c).

One small group of myofibroblasts, the MKI16[+] cells, expressed *MKi67*, indicating that the cell population is undergoing mitosis (Supplementary Fig. 5c, d). One small group of myofibroblasts, RGS5[+]DCN[+] myofibroblasts, expressed *DCN* and *CXCL14* and were similar to fibroblasts.

In cancer tissues, all the mucosa-resident myofibroblast populations were identified (Fig. 4, Supplementary Fig. 5a, b, c). α-SMA staining showed that the positive myofibroblasts were abundant around tumor cells (Fig. 6b to g). The ratio of CD36[hi] myofibroblasts dramatically decreased (Supplementary Fig. 5b, d), which was consistent with previous studies[27]. The deficiency of CD36[hi] cells may be one of the key factors for the functional alteration of CAFs. On the other hand, the proportion of ACAT2[low]CD248[hi] myofibroblasts was highly increased. The fraction of RGS5[+]MKI16[+] myofibroblasts also showed an upward tendency. The ratio of the ACTA2[hi]S100A4[hi] population was unchanged. Trajectory analysis (Supplementary Fig. 5e, f) and the gene expression patterns in single cells revealed that the expression of genes related to RGS5[+] myofibroblasts' structures and functions were not dramatically altered in colon cancer tissues in comparison to human colon mucosa.

**Fibroblast populations that dominantly produce extracellular matrix in colon mucosa and colon cancer tissues.** Osteoglycin (OGN), also known as mimecan, a proteoglycan with great structural and functional diversity in the extracellular matrix[28], was detected in three fibroblast types (Fig. 1e, f), OGN[+]PI16[+], OGN[+]IGF1[+] and OGN[+]KCNN3[+] fibroblasts, in human colon

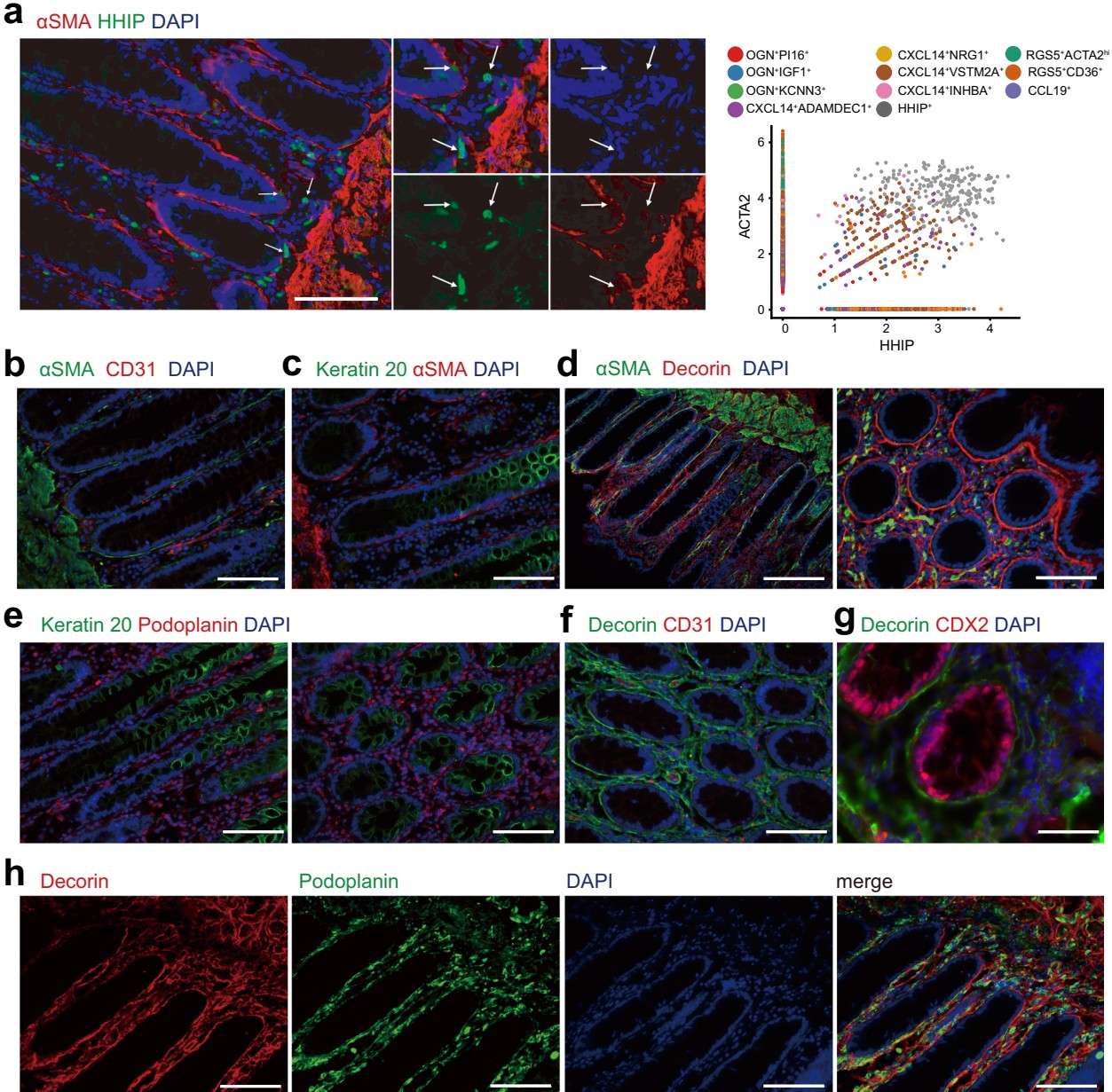

**Fig. 5 Immunofluorescence staining of fibroblasts and myofibroblasts in human colonic mucosa. a** Left: HHIP+ fibroblasts in normal mucosa labeled by α-SMA (red) and HHIP (green). Arrows point at the cells expressing both markers. Scale bar indicates 100 μm. Right: The correlation between *ACTA2* and *HHIP* expression in each cell. Fibroblasts in the HHIP+ cluster were double positive for both ACTA2 and HHIP. **b** Double labeling for α-SMA (green) and CD31 (red). Scale bar indicates 100 μm. **c** Double labeling for α-SMA (red) and Keratin 20 (green). Scale bar indicates 100 μm. **d** Double labeling for fibroblast markers α-SMA (green) and Decorin (red) in the colonic mucosa. Vertical section (left) and cross section (right) of the colonic mucosa. Scale bar indicates 200 μm (left) and 100 μm (right). **e** Double labeling for Keratin 20 (green) and Podoplanin (red). Vertical section (left) and cross section (right) of the colonic mucosa. Scale bar indicates 100 μm. **f** Double labeling for Decorin (green) and CD31(red). Scale bar indicates 100 μm. **g** Double labeling for Decorin (green) and CDX2(red) in the colonic mucosa. Scale bar indicates 50 μm. **h** Double labeling for Decorin (red) and Podoplanin (green). Each fluorescent channel was displayed separately. Scale bar indicates 100 μm. DAPI: 4',6-diamidino-2-phenylindole, blue-fluorescent labeling nuclei; α-SMA, α-smooth muscle actin, encoded by ACTA2 gene; Decorin, encoded by DCN gene. Podoplanin encoded by PDPN. HHIP, Hedgehog interacting protein, encoded by HHIP gene. Keratin 20, labeling mature enterocytes. CDX2, Caudal Type Homeobox 2, an intestinal specific transcription factor labeling intestinal epithelial and cancer cell. CD31, Platelet and Endothelial Cell Adhesion Molecule 1, labeling endothelial cells. CD45, Protein Tyrosine Phosphatase Receptor Type C, labeling immune cells.

mucosa. OGN+PI16+ fibroblasts also expressed high levels of Progranulin (GRN) (Fig. 2c), which protects tissues against harmful inflammation and promotes tissue regeneration[29]. Mainly, the three types of fibroblasts produced collagen, a large fraction of extracellular matrix proteins and extracellular matrix-associated proteins and appeared to be the major cell types that

generates the extracellular matrix components of the connective tissue in human colon mucosa (Fig. 2a, b). Specifically, OGN+ PI16+ fibroblasts expressed high levels of the extracellular matrix protein decorin (DCN) (Fig. 2b). DCN staining showed that DCN was present in the extracellular matrix encircling the colonic crypts and the blood vessels (Fig. 5e, f, g, h). The subepithelial

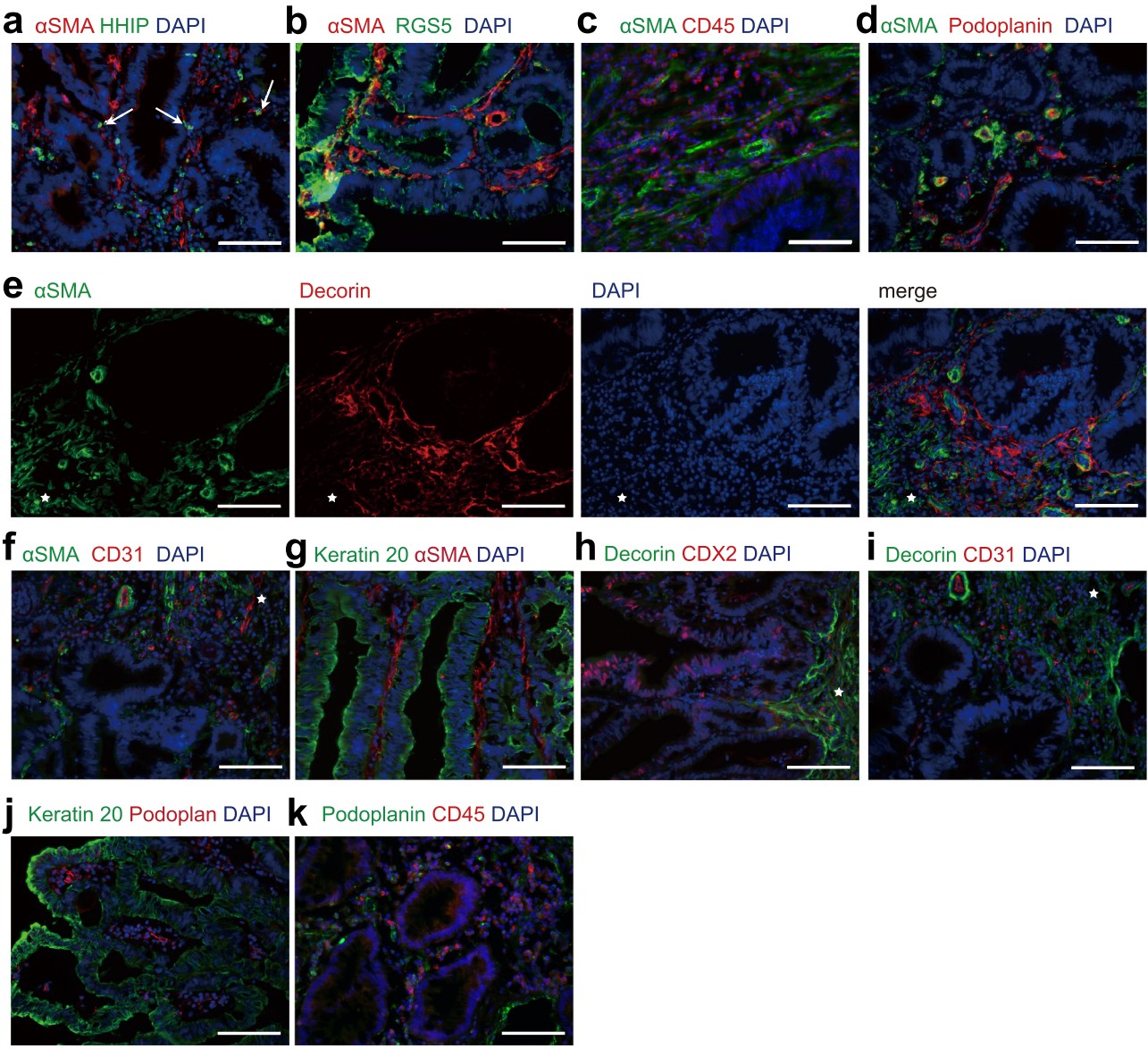

**Fig. 6 Immunofluorescence staining of fibroblasts and myofibroblasts in human colon cancer tissues. a** HHIP$^+$ fibroblasts in CRC colon cancer tissues labeled with αSMA (red) and HHIP (green). Arrows point at the cells expressing both markers. Scale bar indicates 100 μm. **b** Double labeling for α-SMA (red) and RGS5 (green) in colon cancer tissues. Scale bar indicates 100 μm. **c** Double labeling for α-SMA (green) and CD45 (red) in colon cancer tissues. Scale bar indicates 100 μm. **d** Double labeling for α-SMA (green) and Podoplanin (red) in colon cancer tissues. Scale bar indicates 100 μm. **e** Double labeling for α-SMA (green) and Decorin (red) in colon cancer tissues. Each fluorescent channel was displayed separately. **f** Double labeling for α-SMA (green) and CD31 (red) in colon cancer tissues. **g** Double labeling for Keratin 20 (green) and α-SMA (red) in colon cancer tissues. **h** Double labeling for Decorin (green) and CDX2 (red) in colon cancer tissues. **i** Double labeling for Decorin (green) and CD31 (red) in colon cancer tissues. **a–e** The star indicates submucosa adjacent to cancer cells. Scale bar indicates 100 μm. **j** Double labeling for Keratin 20 (green) and Podoplanin (red) in colon cancer tissues. Scale bar indicates 200 μm. **k** Double labeling for Podoplanin (green) and CD45 (red) in colon cancer tissues. Scale bar indicates 100 μm.

myofibroblasts were embedded in the DCN-containing extracellular matrix (Fig. 5g, Supplementary Fig. 6a).

The gene expression patterns showed that the OGN$^+$IGF1$^+$ fibroblasts exhibited similar functions to the OGN$^+$PI16$^+$ fibroblasts in the generation of collagen and extracellular matrix components (Fig. 2a, b). OGN$^+$IGF1$^+$ fibroblasts specifically expressed IGFBP-6 (Fig. 2c), which inhibits the action of insulin-like growth factors (IGFs), mostly IGF2, to modulate cell proliferation, differentiation, migration, survival, apoptosis, angiogenesis and immunoreaction[30]. Notably, the OGN$^+$IGF1$^+$ fibroblasts specifically expressed IGF1 (Fig. 1e). Combined with the fact that OGN$^+$PI16$^+$ fibroblasts secrete IGFBP-6, which mostly inhibits the activity of IGF2, these results suggest that both

types of fibroblasts might promote the reaction of the IGF1 signaling pathway in human colon mucosa.

The third type of OGN$^+$ fibroblast that produced significant extracellular matrix components with distinct patterns was OGN$^+$KCNN3$^+$fibroblasts (Fig. 1f, Fig. 2a, b). This type of fibroblast expressed high levels of *PDGFRA* (Fig. 1f). OGN$^+$KCNN3$^+$ fibroblasts specifically produced the ECM components THBS4 and tenascin-C (TNC) (Fig. 2a, b). THBS4 has been shown to reduce fibrosis and collagen production[31]. TNC modulates cell adhesion to other matrix components and has been shown to block fibronectin-mediated cell adhesion[32].

In colon cancer tissues, the three types of OGN$^+$ fibroblasts were highly decreased or diminished (Fig. 4a, b). Since *DCN* was

identified dominantly to be expressing in OGN$^+$ fibroblasts, consistently, DCN staining was also dramatically decreased and did not encircle any cancer cells in colon cancer tissues (Fig. 6e, h, i).

**Fibroblast populations that preponderantly secret factors in human colon mucosa and colon cancer tissues.** Almost all fibroblasts were identified to express factors including chemokines, chemokine ligands, Wnt ligands and BMPs, in the human colon mucosa. However, the subtypes of CXCL14$^+$ fibroblasts and myofibroblasts were identified to mainly generate secreting factors. Among them, CXCL14$^+$ADAMDEC$^+$ fibroblasts predominantly produced more than 7 chemokines and chemokine ligands (Fig. 2c). This type of fibroblast expressed high levels of the Disintegrin Metalloprotease ADAM-like Decysin-1 (ADAMDEC1) (Fig. 1e, Fig. 2d). ADAMDEC1 is an orphan ADAM-like metalloprotease and is believed closely to associate with inflammation[33]. CXCL14$^+$ADAMDEC$^+$ fibroblasts significantly secreted the chemokines CXCL1, 12, and 14 and the chemokine ligands CCL2, 8, 11 and 13 (Fig. 2c). These chemokines and chemokine ligands have been shown to modulate inflammatory reactions in different ways. Thus, gene expression profiles indicate that a major function of CXCL14$^+$ADAMDEC$^+$ fibroblasts is to modulate inflammation in human colon mucosa.

Fibroblasts in murine colon mucosa secrete ligands to modulate the activities of Wnt or BMP signaling pathways[9]. Thus, we paid close attention to identify the types of fibroblasts that expressed BMPs or Wnt ligands. Three types of such fibroblasts were clustered together among the fibroblast subsets in human colon mucosa. One type of fibroblast, CXCL14$^+$NRG1$^+$ fibroblast, was found to secrete BMP2, BMP4, BMP5, and the noncanonical WNT ligand WNT5A (Fig. 2c). This type of fibroblasts also expressed NRG1, VEGFA, CSF1 and IGFBP3. All the factors secreted by CXCL14$^+$NRG1$^+$ fibroblasts have been reported to exert multifaceted protective effects on epithelial cells, vascular cells, and neural cells and have been shown to have anti-inflammatory effects that ameliorate the disruption of colon mucosa homeostasis, suggesting that CXCL14$^+$NRG1$^+$ fibroblasts may maintain cellular homeostasis in human colon mucosa. CXCL14$^+$VSTM2A$^+$ fibroblasts were similar to CXCL14$^+$NRG1$^+$ fibroblasts in gene expression patterns (Fig. 2). They expressed higher levels of BMP4, PDGFD, FGF9, V-Set, Transmembrane Domain Containing 2 A (VSTM2A), ALK, and LTK Ligand 2 (ALKAL2), suggesting the population of fibroblasts may be involved in the regulation of cell proliferation and transformation.

The other type of fibroblasts in human colon mucosa, CXCL14$^+$INHBA$^+$ fibroblasts (Fig. 1), exhibited substantial activation of gene expression and secreted BMP1, BMP2, BMP4, the noncanonical WNT ligand WNT5A, the canonical WNT ligands WNT2 and WNT4, VEGFA, and INHBA (Fig. 2c). These fibroblasts also expressed the cytokines and chemokines IL32, CXCL1, CXCL2, CXCL8, and CXCL14 and produced extracellular matrix proteins, including collagen (Fig. 2a, b, c). The gene profiling results showed that the gene expression in CXCL14$^+$INHBA$^+$ fibroblasts was highly activated and that this type of fibroblasts was highly similar to the activated fibroblasts identified previously[34].

Another interesting type is the CCL19$^+$ fibroblasts, previously referred to as antigen-presenting fibroblasts, was identified as an inflammatory modulator[24]. This group of cells expressed only a small amount of extracellular matrix-associated glycoproteins and collagen, and specifically expressed the cytokines C-C Motif Chemokine Ligand 19/21 (CCL19/21) and TNF Superfamily Member 13b (TNFSF13B) (Fig. 2c). The population was closer to the OGN$^+$ fibroblast on the trajectory analysis dimensionality

reduction graph (Fig. 3c), but it expressed lower level of *OGN* and higher level of *CXCL14* and *ADAMDEC1* (Fig. 1e, f) and was grouped into CXCL14$^+$ fibroblasts. This population of fibroblasts expressed high level of *CD74*, an MHC class II chaperone involved in non-MHC II protein trafficking and the modulation of T-cell and B-cell development, dendritic cell motility, macrophage inflammation and thymic selection[35].

Detailed analysis of integrated colon mucosa with colon cancer tissues revealed that the factor secreting fibroblasts were divided into more subpopulations and their compositions were significantly different between samples from colon mucosa and from colon cancer tissues (Supplementary Fig. 7a, b, Supplementary Data 8, 9). CXCL14$^+$ADAMDEC1$^+$ fibroblast can be further divided into three subgroups (ADAMDEC1$^+$CCL13$^{hi}$, ADAMDEC1$^+$FABP4$^{hi}$, ADAMDEC1$^+$SFRP1$^{hi}$), all of which were significantly reduced in colon cancer tissue. CXCL14$^+$INHBA$^+$ fibroblasts had four subgroups that were highly expressed *SFRP2*, *ACTG2*, *MMP1*, *FLT1* respectively (Supplementary Fig. 7c, d). In cancer tissues, all subgroups of CXCL14$^+$INHBA$^+$ fibroblasts were highly increased (Fig. 4b, c, Supplementary Fig. 7d). In comparison, the proportions of CXCL14$^+$VSTM2A$^+$ fibroblasts were greatly decreased. In addition, at the high resolution of the detailed analysis, we noticed a IGHA1$^+$ subpopulation of fibroblasts which expressed *IGHA1*, *SRGN* and *CCL4* (Supplementary Fig. 7c, d). Another very small subgroup was RGS5$^+$ fibroblasts, which had similar expression profiles with CXCL14$^+$INHBA$^+$ fibroblast and highly expressed RGS5$^+$ myofibroblast markers.

The gene expression profiles revealed that CXCL14$^+$INHBA$^+$ fibroblasts exhibited altered gene expression patterns but did not essentially change their functions in cancer tissues (Fig. 4c, d, Supplementary Fig. 7d). Majorities of hallmark gene sets, especially TNFα signaling and NFκB pathways, were upregulated in the INHBA$^+$MMP1$^{hi}$, INHBA$^+$FLT1$^{hi}$, and RGS5$^+$ subpopulations (Supplementary Fig. 7g). In addition, the INHBA$^+$ fibroblast expressed a higher number of genes (Supplementary Fig. 7h), which indicated that cells were in an activated statue. Correspondingly, the transcriptional factors, which modulate the expression of genes involved in the active gene expression, were identified in the CXCL14$^+$INHBA$^+$ fibroblasts (Supplementary Fig. 7i). The transmembrane mucin-like protein Podoplanin (PDPN) was identified as a marker of fibroblasts in human colon mucosa (Fig. 1d, Fig. 5e, h, Supplementary Fig. 6b, c). In our integrated analysis of human colon mucosa and tumor fibroblasts, *PDPN* was prominently expressing in CXCL14$^+$INHBA$^+$ fibroblasts (Fig. 1d, Fig. 4d). Immunofluorescence staining for podoplanin indicated that CXCL14$^+$INHBA$^+$ fibroblasts were abundantly presented in the intratumoral stroma (Fig. 6j, k).

In summary, we identified 11 types of fibroblasts and myofibroblasts, including OGN$^+$PI16$^+$, OGN$^+$IGF1$^+$, OGN$^+$KCNN3$^+$, CXCL14$^+$ADAMDEC1$^+$, CXCL14$^+$NRG1$^+$, CXCL14$^+$VSTM2A$^+$, CXCL14$^+$INHBA$^+$, HHIP$^+$, RGS5$^+$ACTA2$^{hi}$, RGS5$^+$CD36$^+$, and CCL19$^+$ fibroblasts and myofibroblasts, which exist in human colon mucosa and in colon cancer tissues (Fig. 7).

## Discussion

Qian, et al described KCNN3$^+$, ADAMDEC1$^+$, and SOX6$^+$ fibroblasts and MYH11$^+$ myofibroblast in colorectum[36]. Lee, et al also discovered ADAMDEC1$^+$, BMP5$^+$ (NRG1$^+$), OGN$^+$ PI16$^+$ fibroblast subpopulations in colon mucosa and colorectal cancer tissues via single-cell dataset analysis[37]. Consistent with previous findings, we identified that all of CXCL14$^+$NRG1$^+$, CXCL14$^+$VSTM2A$^+$ and CXCL14$^+$INHBA$^+$ fibroblasts expressed SOX6. The previously reported fibroblast populations were

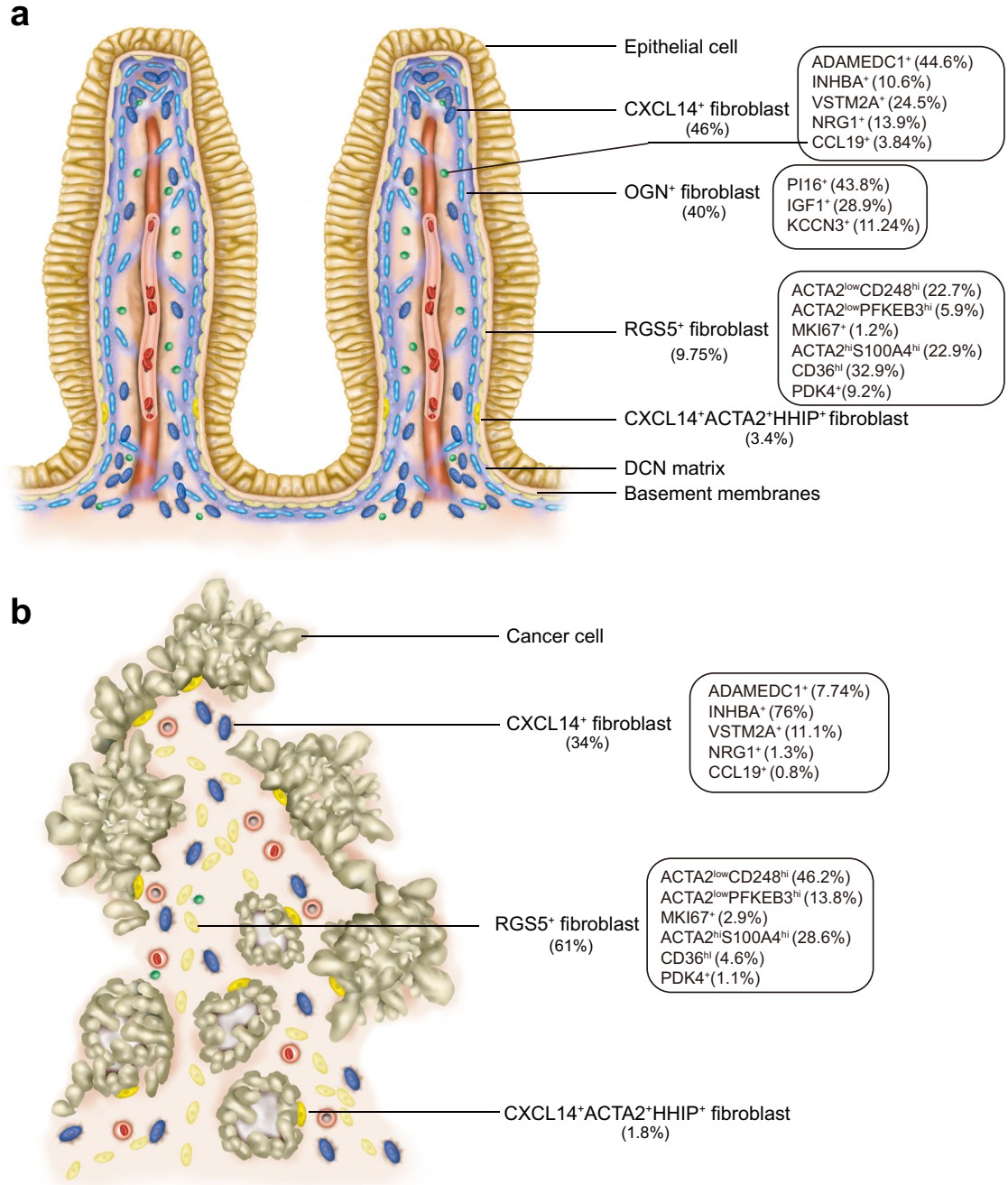

**Fig. 7 Fibroblastic populations in human colon mucosa and colon cancer tissues.** Schematic illustration of the composition and location of types of fibroblasts in **a** normal human colon mucosa and **b** colon cancer tissues. The cell composition ratio of the subpopulation is indicated in the rectangular box. The rest of the cell types are omitted in the schematic illustration.

included in our data. In addition, we provide more detailed information and subpopulations of fibroblasts and myofibroblasts and extend the cellular atlas in the stromal compartment in colon mucosa and in colon cancer tissues.

Previous observations have demonstrated that the intestinal lamina propria consists of layers of extracellular matrix and morphologically distinct fibroblast-like cells along the crypt axis[38]. Intestinal epithelial cells are situated directly above the basement membrane. Under the basement membrane lies a layer of fibroblast-like cells, which have been identified as myofibroblasts, encircling the intestinal crypts[39]. Our results are consistent with the observations and provide more detailed cellular identities of the myofibroblasts. In addition, we identify that a small

fraction of subepithelial myofibroblasts, HHIP[+] myofibroblasts, are located close to the lower part of the colon villi, where colon epithelial stem cells undergo differentiation and proliferation, adjacent to the epithelial cells in the human colon mucosa. These myofibroblasts secrete the canonical WNT ligand WNT2B and a soluble BMP and Wnt antagonist SOSTDC1[40]. The canonical Wnt ligand WNT2B has been shown to play an important role in maintaining intestinal stem cells, and BMP pathways act to induce epithelial differentiation[41,42]. SOSTDC1 is a dual antagonist of the Wnt/β-catenin and BMP signaling pathways, acting to balance canonical Wnt signaling and BMP signaling[40]. Previous observations indicate that canonical WNT signaling and BMP signaling interact to create optimal signaling gradients for

intestinal stem cell maintenance and proliferation[42,43]. Thus, the results suggest that HHIP+ myofibroblasts might act as modulators of intestinal stem cell maintenance and differentiation in human colon mucosa.

The gene expression patterns revealed that RGS5+CD36+ myofibroblasts actively express genes involved in several stimuli-responsive pathways, including cytokine response pathways, IL-17 and TNF pathways and STAT pathways, and might act as contractile cells in response to stimulation in human colon mucosa. RGS5+CD36[low] myofibroblasts have been shown to produce many kinds of collagen and are likely to produce extracellular matrix components in addition to contraction in connective tissue. RGS5+ACTA2[hi] myofibroblasts have been shown to predominantly express many components involved in contractile apparatuses and might be the main cells involved in contraction in connective tissue in human colon mucosa. Our data also reveal mitotic myofibroblasts, RGS+MKI16+ myofibroblasts, fibroblast-like myofibroblasts, and RGS5[low]DCN+ myofibroblasts. Thus, the subepithelial myofibroblast layer of human colon mucosa consists of epithelial cell-modulating myofibroblasts, stimuli-responsive myofibroblasts, proliferating myofibroblasts, fibroblast-like myofibroblasts, matrix producing myofibroblasts and contractile myofibroblasts.

PI16-expressing fibroblasts have been shown to reside among a broad range of tissues[44]. We also identified a type of fibroblasts, OGN+PI16+ fibroblasts, with PI16 expression in human colon mucosa. The gene expression analysis shows that OGN+PI16+ fibroblasts remain at steady state and are the major cells that produce collagen and extracellular matrices in human colon mucosa. Immunostaining revealed that DCN, a small leucine-rich proteoglycan that is produced at high levels by OGN+PI16+ fibroblasts, is present in the extracellular matrix encircling colon crypts. DCN acts as a structural molecule for collagen fibrillogenesis, in organizing the extracellular matrix to maintain the structural integrity of tissues, and as a secreted signaling molecule involved in regulating physiological and pathological processes including cell growth, morphogenesis and immunity[45]. The results indicate that OGN+PI16+ fibroblasts act mainly as steady-state fibroblasts that generate the extracellular matrix components that maintain the structure of the colon lamina propria. We identified OGN+KCNN3+ fibroblasts as one type of fibroblast that significantly produced extracellular matrix components with distinct patterns. The OGN+KCNN3+ fibroblasts were found to produce some specific extracellular matrix components to reduce fibrosis and collagen production, modulate cell adhesion to other matrix components and block fibronectin-mediated cell adhesion. In addition, OGN+KCNN3+ fibroblasts were identified in only two individual samples. These results suggest that OGN+KCNN3+ fibroblasts might respond to reinforce loose connective tissues and are highly divergent in human colon mucosa.

We identified several fibroblast populations that secrete factors including chemokines, chemokine ligands, Wnt ligands and BMPs in human colon mucosa. One type of fibroblast, CXCL14+NRG1+ fibroblasts, produces mainly the noncanonical WNT ligand WNT5A and BMPs. WNT ligands and BMPs are involved in the differentiation and proliferation of intestinal epithelial cells. Thus, this type of fibroblasts might participate in regulating the integrity and homeostasis of human colon mucosa. One type of fibroblasts, CXCL14+ADAMDEC+ fibroblasts, expresses high levels of chemokine and chemokine ligands and is highly similar to the inflammatory fibroblasts identified previously[18]. The gene expression patterns reveal that the third type of secreting fibroblasts, the CXCL14+INHBA+ fibroblast, exhibits high gene activities. The fibroblasts have also been identified to express a wide range of genes, including chemokines, chemokine ligands, canonical WNT ligands, noncanonical WNT ligands, BMPs and extracellular matrix proteins. The features of this fibroblast type are similar to those of the activated fibroblasts identified in previous observations[11,35].

We failed to reveal additional types of fibroblasts and myofibroblasts in colon cancer tissues in comparison to human colon mucosa. However, the OGN+ matrix-producing fibroblasts were dramatically decreased in cancer tissues. We also found that the ratio of fibroblasts that mainly produce chemokines and chemokine ligands is decreased in cancer tissues. Instead, activated fibroblasts have been identified to be highly expanded in colon cancer tissues. In addition, collagen-producing myofibroblasts are greatly increased in cancer tissues. Increased fibroblasts and myofibroblasts have been shown to enhance the expression activities of chemokines, chemokine ligands and many kinds of extracellular matrix proteins, including collagens. Therefore, one role of both increased fibroblasts and myofibroblasts might be to compensate for the decreased matrix producing functions and inflammatory modulating functions of the fibroblasts in cancer tissues. We also identified that the subepithelial myofibroblasts do not encircle epithelial cancer cells in colon cancer tissues. The gene expression patterns show that gene activities in canonical WNT ligand WNT2B-producing myofibroblasts, which are located adjacent to cancer cells, are also altered. The ratio of stimuli-responsive myofibroblasts is significantly decreased in cancer tissues. Multiple studies have demonstrated that ACTA2+ myofibroblast is significantly increased and heterogeneous in tumors[37,46,47], which in line with our findings. All of these results suggest that myofibroblasts might respond to changing environments by adjusting their population, distribution and gene activation in cancer tissues. Taken together, our data suggest that changes in the subtype compositions and functions of fibroblast and myofibroblast populations in cancer might be altered by and for the initiation and progression of cancer cells.

Together, our data systematically delineate the landscape of fibroblasts and myofibroblasts and highlight the cellular heterogeneity of stromal cells in human colon mucosa and in colon cancer tissues. Future cellular approaches should be developed to investigate fibroblasts and myofibroblasts to understand how the integrity and homeostasis of colon mucosa are maintained under diverse physiological and pathological conditions and how colon cancer cells survive and progress.

## Methods

**Human sample collection**. We recruited 12 colorectal cancer patients who underwent radical colectomy. The Ethics Committee of West China Hospital, Sichuan University approved this study and protocols. All patients included in our study were informed by their referring oncologist that their biological samples could be used for research purposes and gave their verbal informed consent. Normal tissues were sampled from sites at least 10 cm away from cancer tissue. Colon cancer tissues and distant normal colonic mucosa were immediately collected after colectomy and transferred into ice-cold 1640 medium (2% BSA, 5% FBS) or DMEM/F12 medium and then transferred to the laboratory on ice. A portion of the tissue was used to prepare a single-cell library, and the remaining portion was processed for staining experiments.

**Immunofluorescence staining**. Tissues were fixed with 4% paraformaldehyde for 1 hour and then dehydrated with 30% sucrose overnight at 4 °C. Thereafter tissues were embedded in O.C.T. Compound (SAKURA Tissue-Tek®) and sectioned to 4 μm on a Slee MEV+ cryostat. Fixed tissue and sections were stored at -80°. For immunofluorescence staining, sections were thawed and rinsed with PBS 3 times for 5 min. After rinsing, sections were permeabilized with 0.1% Triton X-100 for 10 min and blocked with 4% BSA for 1 hour. Thereafter, sections were sequentially incubated with primary antibodies (anti-RGS5 (Proteintech, 11590-1-AP), anti-decorin (Proteintech, 14667-1-AP), anti-podoplanin/gp36 (Abcam, ab10288), anti-CD36 (Beyotime, AF6414), anti-actin (C-2) (SANTA, sc-8432), anti-CDX2 (C-2) (DAKO, M3636), anti-CD31 (PECAM-1)(89C2) (Cell Signaling, #3528), anti-Cytokeratin 20 (Abcam, ab97511), anti-CD45 (proteintech, 80297-1-RR), and anti-HHIP (5D11) (SANTA, sc-293265)) and the corresponding fluorescently conjugated secondary antibodies. Finally, sections were incubated with 4,6-diamidino-2-phenylindole (DAPI, diluted 1:3000 with blocking solution). Micrographs were

acquired using a ZEISS Scope A1 microscope and processed with ZEISS ZEN software.

**Single cell isolation and fluorescence activated cell sorting of fibroblasts**. To obtain single-cell suspensions, samples were cut into small pieces (approximately 1 mm³) and digested with a defined cell dissociation solution containing collagenase I (A004194), collagenase II (A004174), collagenase IV (A004186), dispase (A002100) and DNase I (CAS 9003-98-9) for 40-60 min at 37 °C. Cells were then filtered through a 40-μm cell strainer and resuspended at a final concentration of approximately 5×10⁶ cells/ml.

Single-cell suspensions were then stained for 20 min at 4 °C with a fluorophore-conjugated antibody mix containing anti-CD326-PE-Cyanine7 (25-9326-42), anti-CD45-BV510 (Cat 563204), and anti-CD31- BB515 (Cat 564630). All antibodies were purchased from BD Bioscience. The stained cells were analyzed and sorted using BD Melody (BD Biosciences) in BD FACSDiva software. DAPI was added just before flow cytometry analysis to exclude dead cells. Cells were first gated based on FSC-A/FSC-H scatters to exclude doublet cells and FSC-A/SSC-A scatters to exclude debris. Dead cells were excluded based on their positive staining for DAPI. Single cells from 9 patients were next sorted by gating on CD326⁻, CD45⁻, and CD31⁻ to remove epithelial (CD326⁺), hematopoietic (CD45⁺), and endothelial (CD31⁺) cells. The cells from the other 3 patients were negatively selected only by CD45. Cells from the negative fraction were then used to prepare a single-cell mRNA sequencing library. The data were analyzed with FlowJo software (v.10.8).

**Library construction, sequencing and data processing**. Library construction of CD326CD45⁻CD31⁻ single cells was performed using the BD Rhapsody™ Whole Transcriptome Analysis (WTA) Amplification Kit following a standard protocol (Single-cell 3' whole transcriptome amplification (WTA) with sample multiplexing kit (SMK) protocol) provided by the manufacturer (BD Biosciences). Sequencing was performed by Novogene Company Limited.

The BD Rhapsody WTA Local bioinformatics pipeline was used to process sequencing data (.fastq.gz files). The local pipeline installation was performed on the Ubuntu 18.04 operating system following the BD Rhapsody bioinformatics user guide. Cell labels and molecular indices were identified, and gene identity was determined by alignment to the GRCm37 Genome Reference (genome fasta and annotation gtf file). Next, we used the RSEC_MolsPerCell.csv file (molecules per gene per cell) of each sample for the following clustering analysis. Another fraction, libraries of CD45⁻ single cells from 3 patients were constructed by GEXSCOPE Single Cell RNA Library Kits.

**Dimension reduction and unsupervised clustering**. All downstream analyses were implemented using R version 4.0.2. Low-quality cells (percent of mitochondrial genes above 20%) were filtered before clustering analysis. Dimension reduction and unsupervised clustering were performed in the R package Seurat version 3[48]. To address individual differences and batch effects, we used the SCTransform function to normalize our data and performed integrated analysis with a series of successive functions: PrepSCTIntegration, FindIntegrationAnchors, and IntegrateData. Next, dimension reduction was performed with principal component analysis (PCA) and uniform manifold approximation and projection (UMAP) with variable genes[49]. Finally, we performed the FindClusters function to cluster cells using the Louvain algorithm. After parameter adjustment, this workflow can correct for batch effects and achieve the appropriate clustering.

**Marker identification and differentially expressed gene (DEG) analysis**. After clustering, the subsequent expression analyses were performed based on normalized data (Seurat "RNA" Assay). We used the FindAllMarkers function (logfc.threshold = 0.5) to identify cluster markers and used FindMarkers to find differentially expressed genes between groups or clusters. Both functions used Wilcoxon rank sum test. The DEGs used for display or enrichment analysis had adjusted p values<0.05. P value adjustment was performed using Bonferroni correction based on the total number of genes in the dataset. Gene expression was visualized using functions provided in Seurat: FeaturePlot, DotPlot, and VlnPlot. Heatmaps showing the avg_log2FC of DEGs were built with the pheatmap R package. Volcano plots were plotted by the EnhancedVolcano R package.

**Lineage trajectory**. Pseudotime analysis was performed with Monocle version 2 to infer the potential lineage differentiation trajectory[25,50]. The newCellDataSet function (lowerDetectionLimit = 0.5) was used to build the object based on the normalized data and metadata of the above Seurat object. Genes used to order cells were picked by the Monocle::differentialGeneTest function (qval < 0.001).

**Enrichment analyses**. Enrichment analyses were performed with the clusterProfiler package in R[51]. For cluster markers and DEGs between normal mucosa and CRC for each cluster, we performed overrepresentation analysis (ORA) to identify the functional profiles. For comparison between two clusters, we performed gene set enrichment analysis (GSEA) to rank the list of DEGs[52]. All enrichment analyses

were based on the Gene Ontology (GO) Knowledgebase and Kyoto Encyclopedia of Genes and Genomes (KEGG) Database. Bar plots and dot plots displaying the enrichment results were generated with the ggplot2 package.

**Statistics and reproducibility**. All statistical analyses and visualization were performed in R (The R Project for Statistical Computing) Version 4.0.2. For most experiments, comparisons were made using a two-tailed, unpaired Student's *t* test. The values shown in each figure represent the means ± s.d. *P* < 0.05 was considered statistically significant.

**Reporting summary**. Further information on research design is available in the Nature Portfolio Reporting Summary linked to this article.

## Data availability

The raw sequence data reported in this paper have been deposited in the Genome Sequence Archive (Genomics, Proteomics & Bioinformatics 2021) in National Genomics Data Center (Nucleic Acids Res 2021), China National Center for Bioinformation / Beijing Institute of Genomics, Chinese Academy of Sciences (GSA-Human: PRJCA007492) that are publicly accessible at https://ngdc.cncb.ac.cn/gsa-human. All source data underlying the graphs and charts presented in the main figures have been presented in Supplementary Data.

## Code availability

No new methods and algorithms were used in this study. The analysis methods and software used in this article are open source. Code is available from the corresponding author upon reasonable request.

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

## Acknowledgements

The Ethics Committee of West China Hospital, Sichuan University approved this study. We thank the participants and their families for their kind cooperation, generosity, and patience. The authors thank Yifan Mo, The Johns Hopkins Hospital, for manuscript editing. This work was supported by the National Nature Science Foundation of China (81972592) and the 1.3.5 project for disciplines of excellence of West China Hospital (ZYGD20007, ZYJC18011).

## Author contributions

S. L.: helped by L. S., Data validation, analysis and interpretation, performed immuno-logical data acquisition, analysis, and interpretation. R. L.: Data validation and analysis performed the experimental design and execution. C. F.: Clinical data collection, experimental design and execution. Y. C., J. Z., W. M., Q. H. and X. L: performed experiments. J. D.: Schematic illustration. F. L. and Y. L.: Resources and clinic data collection and analysis. Z. Z.: Clinic supervision, resources and clinic data collection and analysis. X. M: performed conceptualization and supervision, data validation, analysis and interpretation.

## Competing interests

The authors declare no competing interests.
