## [Peer Review File · Communications Biology]

Reviewers' comments:

Reviewer #1 (Remarks to the Author):

This study reports analyses of single cell RNA seq data from human colon and human colon cancer tissue, focusing on mesenchymal cells obtained following a negative selection procedure where cells positive for CD326, CD45 or CD31 were excluded.

The presented analyses report data from a subset of selected cells which are negative for markers above and also Vimentin-positive and Desmin-negative (line 105-106).

The topic is timely and relevant and the data potentially valuable.

Main figure panels report cell clusters and marker genes for cells from normal colon (Fig. 1), trajectory analyses of these subsets (Fig. 2), some comparative analyses of cells from normal tissue and cancer tissue (Fig. 3), detailed description of expression profiles of the subsets (Fig. 4) and four double stainings performed on normal tissue (Fig. 5) and cancer tissue (Fig. 6).

However, as detailed below the present version fails to convincingly convey a clear set of findings, which are presented in context of other recent publications.

Main points:

1. The organization of the study is very un-satisfactory. The mismatch in order of text and Figs make it very hard to follow and evaluate claims.

For example, Figs 5 A-D occur in the order of line 261, 120, 179 and 211. As for Fig. 6, panels B and C are referred to at lines 287 and 198, respectively.

From the Results section perspective, similar lack of order is occurring. For example, the Results section between lines 176 and 241 presents Main Fig results in the order 1E, 5C, 1E, 5C, 3B, 3C, 6C, 1A, 1F, 5D, 3 and 5.

2. A series of major cell subset identification studies in colon cancer have been published, including Li et al, Nat Gen, 2017; Lee et al, Nat gen, 2020; Pelka et al, Cell, 2021. Of these, only the Pelka study is included in the reference list and referred to very briefly (line 205). The lack of a discussion of the present findings, in context and relationships to these other studies, significantly reduces the possibility to evaluate how these findings go beyond earlier studies.

3. No rationale is provided for restricting analyses to the Vim+/Des- cells following the initial cell sorting.

4. Double stainings are commendable to in situ validate subsets. However, as of now the rationale for marker selection, and which subsets they are expected to validate is not described. Analyses in Fig. 6 of cancer tissue uses four panels where alphaSMA is used together with podoplanin, decorin, HHIP and RGS5, whereas Fig.5 uses three of these combinations and also decorin together with podoplanin.

5. Antibody stainings have the potential to increase understanding of spatial organization. However, for this purpose, analyses would have been much more informative if additional markers for cancer cells and endothelial cells, and possibly immune cells, were included.

5. The terms "fibroblasts", "myofibroblasts" and "pericytes" are used without clear definitions which reduces clarity of study.

Reviewer #2 (Remarks to the Author):

In this manuscript, the authors provide in-depth characterization by scRNA-seq of stromal populations in the normal and cancerous colon. This is an interesting manuscript, since the heterogeneity of stromal cells until very recently has been overlooked, however there are several issues that should be addressed in order to make this a cohesive study. Specifically, these are some of the concerns I find with this work:

- 1) Several recent studies characterizing the stromal compartment of the colon compartment are not cited or discussed. The authors should extensively compare and contrast those published subsets of fibroblasts with the fibroblast populations described in the submitted manuscript. E.g. Qian and colleagues (<https://pubmed.ncbi.nlm.nih.gov/32561858/>) found several populations (i.e. C1_KCNN3, C2_ADAMDEC1, C3_SOX6, C7_MYH11, C8_RGS5 etc) that appear to be similar to subsets in the manuscript. Lee et al (<https://pubmed.ncbi.nlm.nih.gov/32451460/>) also described a number of fibroblast subsets, such as an OGN+ population, that seem to share overlapping gene programs with the stromal cell clusters characterized herein.
- 2) Of the 12 samples subjected to scRNA-seq analysis, 9 were FACS-sorted to select for fibroblasts whereas 3 were unsorted, however I cannot find any analyses that show if there are any differences in specific fibroblast populations between these two approaches. I encourage the authors to show such data. It would also be good to show expression pattern of EPCAM, PECAM1, PTPRC and DES, as a control for that the FACS-enrichment protocol was working as expected.
- 3) Regarding the removal of mucosal plasma cells (sFig. 1B-C), it would be good to know which clusters that were excluded from further analysis? I assume cluster 2,4, and 10 were excluded, but were also 0, 5, 7, 8, 12, and 13 excluded, since they also seem to express immunoglobulin genes at appreciable levels? In addition, PMP22 does not seem to be a very specific marker of glial cells.
- 4) On page 4 the authors state that: "Nine clusters, including SRGN+CD74+, OGN+PI16+, OGN+IGF1+, CXCL14+ADAMDEC+, CXCL14+NRG1+ and CXCL14+INHBA+ fibroblasts and CXCL14+ACTA2+HHIP+, RGS5+ ACTA2hi and RGS5+CD36+ myofibroblasts, were identified in all specimens (Fig. 1A, B, C, sFig. 1D)." However, it is apparent by looking at sFig. 1D that e.g. SRGN+CD74+ cells could not be found in patients 1-3 and 11. Furthermore, the CXCL14+ACTA2+HHIP+ population looks to be missing in the samples from patients 4, and 9-10.
- 5) What is the explanation behind the significant upregulation of CD74 and INHBA, which are two top markers of other subpopulations than the CXCL14+ACTA2+HHIP+ subset, and the downregulation of SOSTDC1 and NPNT (two top markers of the original CXCL14+ACTA2+HHIP+ cluster) when looking at the volcano plot showing differentially expressed genes in the CXCL14+ACTA2+HHIP+ cluster between mucosa and colon cancer tissues?
- 6) On page 5 (rows 205-209), the authors make statements regarding markers of pericytes, however except for PDGFRB (Fig. 1F) and DESMIN (I cannot find the expression shown for this gene), it would be appreciated if there were markers (such as CD248, CSPG4, KCNJ8, and ANPEP) of this cell type collected in a few violin plots.
- 7) The two MKI67+ populations that appear in Fig. 3A (they are also covered in more detail in Fig. 3D-E) are never mentioned or discussed in the text.
- 8) In sFig. 1E, why does the CXCL14+INHBA+ cluster have a much higher number of genes compared to all other clusters?
- 9) The scales of many figures lack a unit.
- 10) Cluster number should be added to Fig. 1A to make it easier to discern.
- 11) The resolution of the IF images needs to be increased, making them hard to interpret in their current form.

Reviewer #3 (Remarks to the Author):

This is an important and well written manuscript on the heterogeneity of fibroblasts in normal colon and colorectal cancer (CRC). The single cell data analyzing about 9000 fibroblasts from normal colon and about the same number of CRC are convincing and significantly contribute to the understanding of fibroblast diversity and their potential function in normal gut homeostasis and in CRC.

Minor points:

1.) This is not the first study dealing with fibroblast heterogeneity by single cell sequencing, thus the following papers are mandatory to mention and the results provided in this manuscript should be discussed in the light of the already published data:

Dalerba et al 2011

Qi et al 2021

Kim 2021

Zhang 2020

Li 2017

Qian 2020

...it could be possible that I missed one or two of these papers, so please carefully recherche whether there is other literature out on this issue.

2.) Include one figure (maybe in the supplement) showing expression profiles of so far defined fibroblast markers and CAF markers across the subtypes in normal colon and CRC or used in mouse cre experiments. There are a lot of reviews dealing with the so far established markers (e.g. COL1A1, COL1A2, COL6A1, FAP, FSP1, PDPLN, THY1, VIM, ACTA2, TAGLN, PDGRFA/B, ...), thus, a concise display of these markers (and some more to be found in these reviews) in one comparison in normal colon and in CRC would help the community to progress further and may be of valuable help for functional experiments.

3.) Comment on telocytes (there are several papers on telocytes in the gut) and show their markers in the subsets, maybe it is possible to identify one subset as telocyte subset. This would further contribute to a better understanding of heterogeneity of connective tissue cells for the entire community.

4.) Provide a list of other differentially expressed genes in the identified 9 subtypes, not only the signature genes if this is possible. A set of 20-30 genes in each group even when not fully different in all subgroups might help other researches to better define their subsets or cell which they are working with.

Response to reviewer comments:

Reviewer #1 (Remarks to the Author):

Main points:

1. The organization of the study is very un-satisfactory. The mismatch in order of text and Figs make it very hard to follow and evaluate claims.

For example, Figs 5 A-D occur in the order of line 261, 120, 179 and 211. As for Fig. 6, panels B and C are referred to at lines 287 and 198, respectively.

From the Results section perspective, similar lack of order is occurring. For example, the Results section between lines 176 and 241 presents Main Fig results in the order 1E, 5C, 1E, 5C, 3B, 3C, 6C, 1A, 1F, 5D, 3 and 5.

R: Thank you for your correction. We have checked and revised the paper, and the mismatch of text and figures have been corrected.

2. A series of major cell subset identification studies in colon cancer have been published, including Li et al, Nat Gen, 2017; Lee et al, Nat gen, 2020; Pelka et al, Cell, 2021. Of these, only the Pelka study is included in the reference list and referred to very briefly (line 205). The lack of a discussion of the present findings, in context and relationships to these other studies, significantly reduces the possibility to evaluate how these findings go beyond earlier studies.

R: We have included more discussion of previous related research and our findings in the discussion section. Li et al found two distinct subtypes of cancer-associated fibroblasts (CAFs) PDPN /ACTA2. Lee et al identified several fibroblast and myofibroblast subtypes (S1, S2, S3, and MF, MF2, MF3 ,MF4. The characteristics of some populations are similar to our study, such as the ADAMDEC1⁺ S1, BMP5⁺ S2 (CXCL14⁺NRG1⁺ fibroblast) and OGN⁺ fibroblast. our study performed more detailed analysis with enriched fibroblasts and provide an extended cellular atlas of fibroblastic populations in normal colon mucosa and colon cancer tissue. The related papers are now included in our reference list.

3. No rationale is provided for restricting analyses to the Vim⁺/Des⁻ cells following the initial cell sorting.

R: In the revised manuscript, we have adopted a single-cell clustering method to obtain fibroblastic populations instead of direct negative marker filtering, the specific process was shown in sFig. 1D~E. All cell populations included in our analysis expressed Vim, COL1A1 and THY1 (Fig.1D).

4. Double stainings are commendable to in situ validate subsets. However, as of now the rationale for marker selection, and which subsets they are expected to validate is not described. Analyses in Fig. 6 of cancer tissue uses four panels where alphaSMA is

used together with podoplanin, decorin, HHIP and RGS5, whereas Fig.5 uses three of these combinations and also decorin together with podoplanin.

R: In our datasets, DCN is highly expressed in OGN⁺ and CXCL14⁺ADAMDEC1⁺ fibroblasts, but lower in the ACTA2⁺ clusters (Fig. 1D, E, F). Decorin is glycoprotein that constitutes the extracellular matrix. Double staining helps us to better understand the spatial arrangement of cells in normal mucosa and tumors, allowing us to further explore the functions of various subpopulations. PDPN expressed in OGN⁺ and CXCL14⁺ clusters (Fig. 1D), which encoded a membrane glycoprotein. Co-staining Decorin with α SMA and Podoplanin, respectively, allowed us to observe differences in the distribution of PDPN⁺ and ACTA2⁺ cells. We stained the sections of colon mucosa and cancer tissues via the same panels of antibodies and displayed representative results in the manuscript.

5. Antibody stainings have the potential to increase understanding of spatial organization. However, for this purpose, analyses would have been much more informative if additional markers for cancer cells and endothelial cells, and possibly immune cells, were included.

Our study provides a detailed classification of the diversity of fibroblasts by single-cell sequencing analysis and staining of enriched fibroblasts, and describes their distribution and function within tissues. We sectioned serially the samples of human mucosa and cancer tissues and identified the cell types in mucosa and cancer tissues by HE staining the matched sections. Some HE staining sections as following:

colon Mucosa

(cross section)

colon Mucosa

(vertical section)

Colon cancer tissue

Colon cancer tissue

5. The terms “fibroblasts”, “myofibroblasts” and “pericytes” are used without clear definitions which reduces clarity of study.

R: We have added definitions to these terms in the main text.

Reviewer #2 (Remarks to the Author):

Specifically, these are some of the concerns I find with this work:

1) Several recent studies characterizing the stromal compartment of the colon compartment are not cited or discussed. The authors should extensively compare and

contrast those published subsets of fibroblasts with the fibroblast populations described in the submitted manuscript. E.g. Qian and colleagues (<https://pubmed.ncbi.nlm.nih.gov/32561858/>) found several populations (i.e. C1_KCNN3, C2_ADAMDEC1, C3_SOX6, C7_MYH11, C8_RGS5 etc) that appear to be similar to subsets in the manuscript. Lee et al (<https://pubmed.ncbi.nlm.nih.gov/32451460/>) also described a number of fibroblast subsets, such as an OGN+ population, that seem to share overlapping gene programs with the stromal cell clusters characterized herein.

R: We have added a discussion of these two papers to the Discussion section in the main text.

2) Of the 12 samples subjected to scRNA-seq analysis, 9 were FACS-sorted to select for fibroblasts whereas 3 were unsorted, however I cannot find any analyses that show if there are any differences in specific fibroblast populations between these two approaches. I encourage the authors to show such data.

R: We presented the analytical procedures for three pairs of CD45- sorted samples in the supplementary material (sFig1.B,C). The difference between the results of the two experiments is also shown (sFig1, D, top right).

It would also be good to show expression pattern of EPCAM, PECAM1, PTPRC and DES, as a control for that the FACS-enrichment protocol was working as expected.

R: In the revised manuscript, we have adopted a single-cell clustering method to obtain fibroblastic populations instead of direct negative marker filtering, the specific process was shown in sFig. 1D~E. All cell populations included in our analysis expressed Vim, COL1A1 and THY1 (Fig.1D).

After filtering cells by the new method, we found that PECAM1, PTPRC and DES positive cells were very low in the data sets of both experimental methods and did not affect the clustering, while EPCAM positive cells were relatively higher in Singleron GEXSCOPE built dataset.

The proportion of EPCAM-positive cells in the fibroblastic populations isolated from the CD45-sorted three patients' dataset is 25%, which far exceeds the proportion of fibroblast & Epithelial cell doublets that may exist. While the proportion of EPCAM-positive cells in CD45-/CD31-/CD326- sorted dataset is only 1%. At this time, we cannot determine the reason for this discrepancy. We speculate that some fibroblast subgroups will express some EPCAM, and the EPCAM⁺ cells specifically enriched in the CXCL14⁺ VSTM2A⁺ subcluster in the normal mucosa fibroblast.

Compared with the previous analysis, keep this part cells allowed us to separate two subclusters, CXCL14⁺VSTM2A⁺ and CXCL14⁺NRG1⁺. These two groups of cells were grouped together in our original manuscript.

Compare of VSTM2A and NRG1 expression in normal mucosa fibroblastic population

3) Regarding the removal of mucosal plasma cells (sFig. 1B-C), it would be good to know which clusters that were excluded from further analysis? I assume cluster 2,4, and 10 were excluded, but were also 0, 5, 7, 8, 12, and 13 excluded, since they also seem to express immunoglobulin genes at appreciable levels? In addition, PMP22 does not seem to be a very specific marker of glial cells.

R: Mucosal plasma cells and Stromal cell subtypes such as endothelial cell and enteric glial cell also play some significant roles in normal mucosa and tumor in human colon, but this paper only discussed the characteristic and function of fibroblastic populations because of space limit. Clusters that were included in the main analysis and their celltype markers were showed in the revise sFig 1. The enteric glial cells were characterized by expression of CDH19, PLP1, and SOX10 etc.

4) On page 4 the authors state that: "Nine clusters, including SRGN+CD74+, OGN+PI16+, OGN+IGF1+, CXCL14+ADAMDEC+, CXCL14+NRG1+ and CXCL14+INHBA+ fibroblasts and CXCL14+ACTA2+HHIP +, RGS5+ ACTA2hi and RGS5+CD36+ myofibroblasts, were identified in all specimens (Fig. 1A, B, C, sFig. 1D)." However, it is apparent by looking at sFig. 1D that e.g. SRGN+CD74+ cells could not be found in patients 1-3 and 11. Furthermore, the CXCL14+ACTA2+HHIP+ population looks to be missing in the samples from patients 4, and 9-10.

R:

The composition of cells of each cluster in the patient

Virtually all subpopulations were present in multiple patients, but due to the large variability in cell quantity and quality between samples in our dataset, some patients collected fewer cells and therefore did not contain all types.

N3 OGN+KCNN3+ cluster was more in two patients (patient 7 and 9), but less in others. N8 HHIP+ myofibroblast was found in 9 patients. SRGN+CD74+ cluster was removed in the revision.

5) What is the explanation behind the significant upregulation of CD74 and INHBA,

which are two top markers of other subpopulations than the CXCL14+ACTA2+HHIP+ subset, and the downregulation of SOSTDC1 and NPNT (two top markers of the original CXCL14+ACTA2+HHIP+ cluster) when looking at the volcano plot showing differentially expressed genes in the CXCL14+ACTA2+HHIP+ cluster between mucosa and colon cancer tissues?

R: For the HHIP⁺ population, genes such as INHBA and CD74 were up-regulated in tumors, and SOSTDC1 and NPNT were down-regulated in tumors, suggesting that this population has gene expression differences between normal mucosa and tumors that cause functional changes (sFig. 4). The characteristics of high expression of HHIP and ACTG2 in this cluster were unchanged in normal mucosa and colon cancer (Fig. 4), and there was a certain corresponding relationship in spatial distribution (Fig. 5, Fig.6).

6) On page 5 (rows 205-209), the authors make statements regarding markers of pericytes, however except for PDGFRB (Fig. 1F) and DESMIN (I cannot find the expression shown for this gene), it would be appreciated if there were markers (such as CD248, CSPG4, KCNJ8, and ANPEP) of this cell type collected in a few violin plots.

R: Pericytes are often characterized as co-expressing TAGLN, ACTA2, RGS5 and PDGFRB.

Qian et al identified RGS5⁺CD36⁺ stromal cells as pericytes. In our analysis, several RGS5⁺ subpopulations co-expressed these genes, and we could not clearly distinguish between myofibroblast and pericyte at the single-cell RNA sequencing level. On the other way, we found that ACTA2⁺CD36⁺ myofibroblast were present close to crypt epithelial cells (sFig. 2A).

Showing pericyte related marker expression in all fibroblastic clusters:

Showing pericyte related marker expression in RGS5⁺ subpopulations:

7) The two MKI67⁺ populations that appear in Fig. 3A (they are also covered in more detail in Fig. 3D-E) are never mentioned or discussed in the text.

R: In the revised version of our manuscript, the MKI67⁺ cells cannot be clustered individually in the CXCL14⁺ population, and the cell ratio in the RGS5⁺ population is increased in CRC, but since the number of cells is too small to be statistically significant, we also described it at corresponding paragraph in the revised manuscript.

8) In sFig. 1E, why does the CXCL14⁺INHBA⁺ cluster have a much higher number of genes compared to all other clusters?

R: CXCL14⁺INHBA⁺ is an activated subtype of fibroblast.

9) The scales of many figures lack a unit.

R: We have revised figures and labels, and added scale units.

10) Cluster number should be added to Fig. 1A to make it easier to discern.

R: We marked the cluster name and number near the dimensionality reduction map to facilitate the corresponding identification.

11) The resolution of the IF images needs to be increased, making them hard to interpret in their current form.

R: We will submit higher resolution images.

Reviewer #3 (Remarks to the Author):

Minor points:

1.) This is not the first study dealing with fibroblast heterogeneity by single cell sequencing, thus the following papers are mandatory to mention and the results provided in this manuscript should be discussed in the light of the already published data:

Dalerba et al 2011

Qi et al 2021

Kim et al 2021

Zhang 2020

Li 2017

Qian 2020

...it could be possible that I missed one or two of these papers, so please carefully recherche whether there is other literature out on this issue.

R: We have included more discussion of previous related research and our findings in the discussion section and cited some of the articles in the list.

2.) Include one figure (maybe in the supplement) showing expression profiles of so far defined fibroblast markers and CAF markers across the subtypes in normal colon and CRC or used in mouse cre experiments. There are a lot of reviews dealing with the so far established markers (e.g. COL1A1, COL1A2, COL6A1, FAP, FSP1, PDPLN, THY1, VIM, ACTA2, TAGLN, PDGRFA/B, ...), thus, a concise display of

these markers (and some more to be found in these reviews) in one comparison in normal colon and in CRC would help the community to progress further and may be of valuable help for functional experiments.

R: In fact, most of these markers were contained in the dot plot of our figures (Fig. 4C). The expression levels of some genes in our dataset were too low or not distinct between subclusters and were not shown in the main text.

3.) Comment on telocytes (there are several papers on telocytes in the gut) and show their markers in the subsets, maybe it is possible to identify one subset as telocyte subset. This would further contribute to a better understanding of heterogeneity of connective tissue cells for the entire community.

R: Telocytes are mesenchymal cells that close contact with the entire crypt base, and form a subepithelial plexus. Telocytes express high levels of genes in several key signaling pathway including Wnt, SHH, BMP and TGF- β . This type of cells play a very important role to remain proliferation of stem and epithelial renewal.

We collected telocytes marker genes proposed in several papers on telocytes and visualized them in our data.

In our data, the exact telocyte marker FOXL1 and GIL1 expressed in several CXCL14⁺ subpopulations of fibroblast.

References:

Shoshkes-Carmel, M., Wang, Y.J., Wangenstein, K.J. et al. Subepithelial telocytes are an important source of Wnts that supports intestinal crypts. *Nature* 557, 242–246 (2018). <https://doi.org/10.1038/s41586-018-0084-4>

Bahar Halpern K, Massalha H, Zwick RK, Moor AE, Castillo-Azofeifa D, Rozenberg M, Farack L, Egozi A, Miller DR, Averbukh I, Harnik Y, Weinberg-Corem N, de Sauvage FJ, Amit I, Klein OD, Shoshkes-Carmel M, Itzkovitz S. Lgr5⁺ telocytes are a signaling source at the intestinal villus tip. *Nat Commun.* 2020 Apr 22;11(1):1936. doi: 10.1038/s41467-020-15714-x. PMID: 32321913; PMCID: PMC7176679.

Rosa I, Marini M, Manetti M. Telocytes: An Emerging Component of Stem Cell Niche Microenvironment. *Journal of Histochemistry & Cytochemistry.* 2021;69(12):795-818. doi:10.1369/00221554211025489

4.) Provide a list of other differentially expressed genes in the identified 9 subtypes, not only the signature genes if this is possible. A set of 20-30 genes in each group even when not fully different in all subgroups might help other researchers to better define their subsets or cell which they are working with.

R: Cluster marker gene table would be provided in the supplementary material.

Reviewers' comments:

Reviewer #1 (Remarks to the Author):

Authors have revised manuscript regarding some of the earlier main concerns including references to earlier studies.

However, regarding main concerns 4 (in situ validation of subsets) and 5 (characterization of subset localization in relationship to cancer cells, vessels and immune cells) no major additions have been done.

The antibody-based in situ analyses of tentative subsets in CRC remain restricted to what is shown in Fig. 6. The design of these analyses is not sufficient neither for in situ validation of the subsets, or for describing compartmentalization of subsets. Notably, I could not find in the results section any references to Fig. 6A or Fig. 6D.

Regarding subset-validation (main point 4) the reply is difficult to understand and follow. Furthermore, authors suggest that Decorin should mark both the "OGN +" and the "CXCL14 +ADAMDEC1 +" which is not suggested by Fig. 4C, which implies DCN as an OGN subset-marker.

Regarding "compartmentalization" of subset (main point 5), no additional co-stainings using markers for vessels, immune cells or cancer cells are presented. Instead, authors enclose in rebuttal letter a series of microphotographs of HE-stainings without major relevance for the issue.

Reviewer #2 (Remarks to the Author):

This is a revised and improved manuscript. The authors have provided new data and/or information that adequately address issues raised by the referees.

Response to Reviewer' comments

Reviewer #1 (Remarks to the Author):

Authors have revised manuscript regarding some of the earlier main concerns including references to earlier studies.

However, regarding main concerns 4 (in situ validation of subsets) and 5 (characterization of subset localization in relationship to cancer cells, vessels and immune cells) no major additions have been done.

The antibody-based in situ analyses of tentative subsets in CRC remain restricted to what is shown in Fig. 6. The design of these analyses is not sufficient neither for in situ validation of the subsets, or for describing compartmentalization of subsets. Notably, I could not find in the results section any references to Fig. 6A or Fig. 6D.

Regarding subset-validation (main point 4) the reply is difficult to understand and follow. Furthermore, authors suggest that Decorin should mark both the “OGN +” and the “CXCL14 +ADAMDEC1 +” which is not suggested by Fig. 4C, which implies DCN as an OGN subset-marker.

Regarding “compartmentalization” of subset (main point 5), no additional co-stainings using markers for vessels, immune cells or cancer cells are presented. Instead, authors enclose in rebuttal letter a series of microphotographs of HE-stainings without major relevance for the issue.

Response:

Thank greatly the reviewer #1 for the suggestions and comments on our manuscript. According to the reviewer's comments and suggestion, we performed new experiments to address the reviewer's concern.

To verify the results obtained from the single cell sequencing, we employed anti-human CD31, CD45, CDX2, and Cytokeratin 20 antibodies to stain endothelial cells of blood vessels, inflammatory cells and epithelial cells or cancer cells in the tissues of samples and sectioned serially the samples of human mucosa and cancer tissues and stained the matched sections with *hematoxylin-eosin* to identify the epithelial cells or cancer cells, blood vessels, mucosa, submucosa, smooth muscle cells and stromal tissues in the sections of samples. The fibroblasts and myofibroblasts were visualized by the identified antigens detected specifically by available anti-human antibodies.

We examined a panel of antibodies against Decorin (DCN), PDPN, α -SMA, RGS5, CD36, HHIP, OGN, and CXCL14 to detect the localizations of fibroblast and myofibroblast subsets in relationship to epithelial cells, or cancer cells, blood vessels and inflammatory cells. We obtained the specific staining results of the antibodies against DCN, PDPN, α -SMA, RGS5, CD36, HHIP. The antibodies of OGN, and CXCL14 did not specifically stain the tissue sections. Then we co-stained the antibodies of DCN, PDPN, α -SMA, HHIP with CD31, CDX2, Cytokeratin 20 or CD45 antibody. The specific represent results included in Figs 5, 6, supplementary Fig 2. Due to the space, we did not include all of results in Figs. We also did not include the results of

matched sections with *hematoxylin-eosin* staining.

DCN is secreted by the majority of OGN⁺ and CXCL14⁺ Fibroblasts (Fig .4c). However, its expressing level is much higher in OGN⁺ Fibroblast subsets. If the tissue sections were stained with lower concentration of the antibody against DCN, the positive stained fibroblasts were OGN⁺ Fibroblast subsets. We identified OGN⁺ fibroblast subsets greatly reduced in cancer tissues in the single cell sequencing datasets. Consistently, the immunofluorescence staining of DCN was dramatically decreased in cancer tissues. The results are agreed to the reviewer's comment and the DCN could be used as an OGN subset-marker.

We modified the results in the text according to the results obtained from new experiments.